# Machine learning suggests polygenic risk for cognitive dysfunction in amyotrophic lateral sclerosis

Katerina Placek[1] (iD), Michael Benatar[2], Joanne Wuu[2], Evadnie Rampersaud[3], Laura Hennessy[1], Vivianna M Van Deerlin[4], Murray Grossman[1], David J Irwin[1], Lauren Elman[1], Leo McCluskey[1], Colin Quinn[1], Volkan Granit[2], Jeffrey M Statland[5], Ted M Burns[6], John Ravits[7], Andrea Swenson[8], Jon Katz[9], Erik P Pioro[10], Carlayne Jackson[11], James Caress[12], Yuen So[13], Samuel Maiser[14], David Walk[14], Edward B Lee[4], John Q Trojanowski[4], Philip Cook[15], James Gee[15], Jin Sha[16,17], Adam C Naj[4,16,17], Rosa Rademakers[18] (iD), The CReATe Consortium[19,†,§], Wenan Chen[3], Gang Wu[3], J Paul Taylor[3,20] & Corey T McMillan[1,*] (iD)

## Abstract

Amyotrophic lateral sclerosis (ALS) is a multi-system disease characterized primarily by progressive muscle weakness. Cognitive dysfunction is commonly observed in patients; however, factors influencing risk for cognitive dysfunction remain elusive. Using sparse canonical correlation analysis (sCCA), an unsupervised machine-learning technique, we observed that single nucleotide polymorphisms collectively associate with baseline cognitive performance in a large ALS patient cohort ($N = 327$) from the multicenter Clinical Research in ALS and Related Disorders for Therapeutic Development (CReATe) Consortium. We demonstrate that a polygenic risk score derived using sCCA relates to longitudinal cognitive decline in the same cohort and also to *in vivo* cortical thinning in the orbital frontal cortex, anterior cingulate cortex, lateral temporal cortex, premotor cortex, and hippocampus ($N = 90$) as well as *post-mortem* motor cortical neuronal loss ($N = 87$) in independent ALS cohorts from the University of Pennsylvania Integrated Neurodegenerative Disease Biobank. Our findings suggest that common genetic polymorphisms may exert a polygenic contribution to the risk of cortical disease vulnerability and cognitive dysfunction in ALS.

**Keywords** amyotrophic lateral sclerosis; cognition; frontotemporal dementia; machine learning; polygenic score

**Subject Categories** Computational Biology; Genetics, Gene Therapy & Genetic Disease; Neuroscience

1 Department of Neurology, University of Pennsylvania Perelman School of Medicine, Philadelphia, PA, USA
2 Department of Neurology, Leonard M. Miller School of Medicine, University of Miami, Miami, FL, USA
3 Center for Applied Bioinformatics, St. Jude Children's Research Hospital, Memphis, TN, USA
4 Department of Pathology & Laboratory Medicine, University of Pennsylvania Perelman School of Medicine, Philadelphia, PA, USA
5 Department of Neurology, University of Kansas Medical Center, Kansas City, KS, USA
6 Department of Neurology, University of Virginia Health System, Charlottesville, VA, USA
7 Department of Neurosciences, University of California San Diego, San Diego, CA, USA
8 Department of Neurology, University of Iowa, Iowa City, IA, USA
9 Forbes Norris ALS Center, California Pacific Medical Center, San Francisco, CA, USA
10 Department of Neurology, Cleveland Clinic, Cleveland, OH, USA
11 Department of Neurology, University of Texas Health Science Center, San Antonio, TX, USA
12 Department of Neurology, Wake Forest University School of Medicine, Winston-Salem, NC, USA
13 Department of Neurology, Stanford University Medical Center, San Jose, CA, USA
14 Department of Neurology, University of Minnesota Medical Center, Minneapolis, MN, USA
15 Penn Image Computing Science Laboratory (PICSL), Department of Radiology, University of Pennsylvania Perelman School of Medicine, Philadelphia, PA, USA
16 Department of Biostatistics, Epidemiology, and Informatics, University of Pennsylvania Perelman School of Medicine, Philadelphia, PA, USA
17 Penn Neurodegeneration Genomics Center, Department of Pathology and Laboratory Medicine, University of Pennsylvania Perelman School of Medicine, Philadelphia, PA, USA
18 Department of Neuroscience, Mayo Clinic, Jacksonville, FL, USA
19 Rare Diseases Clinical Research Network, National Institutes of Health, Bethesda, MD, USA
20 The Howard Hughes Medical Institute, Chevy Chase, MS, USA
*Corresponding author. Tel: +1 215 614 0987; E-mail: mcmillac@pennmedicine.upenn.edu
†Refer to Appendix S1 for full CReATe Consortium author list
§This article has been contributed to by US Government employees and their work is in the public domain in the USA.

## Introduction

A significant proportion of patients with amyotrophic lateral sclerosis (ALS) manifest impairment in cognition consistent with extra-motor frontal and temporal lobe neurodegeneration, including 14% also diagnosed with frontotemporal dementia (FTD) (Montuschi et al, 2015; Beeldman et al, 2016). Comorbid cognitive dysfunction is a marker of poorer prognosis in this fatal disease and confers risk for more rapid functional decline, shorter survival, and greater caregiver burden (Elamin et al, 2013; Hu et al, 2013; Crockford et al, 2018). While linkage analysis and genome-wide association studies (GWAS) have identified rare causal mutations (Van Deerlin et al, 2008; DeJesus-Hernandez et al, 2011; Renton et al, 2011) and common risk loci (van Es et al, 2009; Diekstra et al, 2014; van Rheenen et al, 2016; Nicolas et al, 2018; Karch et al, 2018) suggesting shared genetic architecture between ALS and FTD, whether and how identified variants relate to phenotypic heterogeneity, including in cognition, remain largely unexplored.

The genetic landscape of ALS is largely characterized by "apparently sporadic" disease occurring in 90% of patients with no known family history of ALS and only a small proportion of approximately 10% of patients having a family history of ALS (Turner et al, 2017). Known pathogenic mutations (e.g., C9ORF72 (Renton et al, 2011; DeJesus-Hernandez et al, 2011), TARDBP (Van Deerlin et al, 2008), FUS (Vance et al, 2009), NEK1 (Kenna et al, 2016), SOD1 (Rosen et al, 1993)) have been identified in many familial cases and in 5–7% of non-familial cases (Umoh et al, 2016); in addition, GWAS have revealed many loci of common genetic variation that confer risk for ALS and FTD. Indeed, recent evidence supports a polygenic contribution to disease risk from common genetic variants (Bandres-Ciga et al, 2019). These include the largest ALS GWAS to-date which newly identified risk variants in the KIF5A gene (Nicolas et al, 2018) and genome-wide conjunction and conditional false discovery rate (FDR) analyses demonstrating shared genetic contributions between ALS and FTD from common single nucleotide polymorphisms (SNPs) at known and novel loci (Karch et al, 2018).

An accumulating body of research suggests that SNPs associated with risk of ALS and FTD demonstrate quantitative-trait modification of patient phenotype. For example, a SNP identified as a risk locus for ALS and FTD was found to contribute to cognitive decline, in vivo cortical degeneration in the prefrontal and temporal cortices, and post-mortem pathologic burden of hyperphosphorylated TAR-DNA binding protein [43 kDa] (TDP-43) in the middle frontal, temporal, and motor cortices (Placek et al, 2019). Another study found that a SNP identified as a risk locus for FTD with underlying TDP-43 pathology was additionally associated with cognition in patients with ALS (Vass et al, 2011). Others have recently demonstrated shared polygenic risk between ALS and other traits (e.g., smoking, education) and diseases (e.g., schizophrenia) (Bandres-Ciga et al, 2019), suggesting that a single variant is unlikely to fully account for observed disease phenotype modification. However, there are presently no published studies evaluating polygenic contribution to cognitive dysfunction in ALS.

Here, we employed an unsupervised machine-learning approach, sparse canonical correlation analysis (sCCA) (Witten & Tibshirani, 2009), to identify and evaluate a potential polygenic contribution to cognitive dysfunction in ALS. sCCA has previously been implemented in many contexts such as genetics (Parkhomenko et al, 2007; Witten & Tibshirani, 2009), neuroimaging-behavior studies (Avants et al, 2010, 2014), and neuroimaging-genetic studies (Hao et al, 2017), including the association of cortical thickness and white matter diffusion to FTD risk SNPs (McMillan et al, 2014). For the first time, we leverage sCCA as a data-driven tool to facilitate generation of a polygenic risk score. Specifically, sCCA can be leveraged to select variants by employing sparsity to identify maximally contributing variants and to assign corresponding weights based on model contribution with minimal a priori assumptions. This contrasts with traditional approaches to constructing polygenic scores that rely on the use of existing GWAS statistics to select variants and assign weights, which can be challenging if the original GWAS statistics are based on case–control associations rather than current neuropsychological outcome of interest.

We used sCCA to derive a polygenic risk score for cognitive dysfunction in a large longitudinal cohort of cognitively well-characterized patients with ALS or a related disorder participating in the Phenotype–Genotype–Biomarker (PGB) study of the Clinical Research in ALS and Related Disorders for Therapeutic Development (CReATe) Consortium. We then evaluated independent neuroimaging and autopsy ALS patient cohorts from the University of Pennsylvania Integrated Neurodegenerative Disease Biobank (UPenn Biobank) (Toledo et al, 2014) to evaluate whether polygenic risk for cognitive dysfunction also relates to in vivo cortical neurodegeneration and ex vivo cortical neuronal loss and TDP-43 pathology. We focused our investigation on SNPs achieving genome-wide significance in the largest published ALS GWAS (Nicolas et al, 2018) and SNPs identified as shared risk loci for both ALS and FTD (Karch et al, 2018). We hypothesized that a sparse multivariate approach would reveal a subset of genetic loci associated with cognitive dysfunction profiles in ALS in a polygenic manner, and that follow-up analyses in independent neuroimaging and autopsy cohorts would converge to characterize quantitative traits associated with polygenic risk from identified loci.

## Results

### Heterogeneity of baseline cognitive and motor phenotype in ALS patients

Smaller-scale studies have shown that ALS patients have impairments in executive function, verbal fluency, and language domains, but with relative sparing of memory and visuospatial function (Crockford et al, 2018). The Edinburgh Cognitive and Behavioral ALS Screen (ECAS) was developed to measure cognitive function minimally confounded by motor disability and includes an "ALS-Specific" score that captures impairments in language, executive function, and verbal fluency domains that are frequently observed in ALS patients, and an "ALS-Non-Specific" score that captures less frequently observed impairments in memory and visuospatial function, in addition to overall performance (ECAS Total score) (Abrahams et al, 2014). To quantify heterogeneity in cognitive dysfunction, we evaluated 327 patients with ALS, ALS with cognitive impairment (ALSci), or a related disorder (ALS-FTD, primary lateral sclerosis (PLS), progressive muscular atrophy

**Table 1. Baseline demographic characteristics of the CReATe PGB cohort.**

| | ALS | ALSci | ALS-FTD | PLS | PMA |
|---|---|---|---|---|---|
| N | 113 | 166 | 13 | 22 | 13 |
| Sex, Male (%) | 61 (54.0) | 102 (61.4) | 11 (84.6) | 11 (50.0) | 8 (61.5) |
| Number of Visits, M (SD) | 3.05 (1.41) | 3.11 (1.34) | 3.00 (1.15) | 2.86 (1.28) | 3.38 (1.45) |
| Age at Symptom Onset, M (SD) | 54.50 (12.91) | 57.56 (12.19) | 64.00 (9.11) | 49.68 (7.39) | 48.08 (15.31) |
| Symptom Onset to Baseline (years), M (SD) | 3.48 (3.73) | 3.67 (5.69) | 3.62 (2.63) | 8.45 (6.12) | 7.77 (7.17) |
| Site of Symptom Onset, N (%) | | | | | |
| Bulbar | 23 (22.5) | 22 (13.7) | 4 (33.3) | 5 (22.7) | – |
| Bulbar and Limb | 4 (3.9) | 3 (1.9) | – | 3 (13.6) | – |
| Bulbar and Other | 3 (2.9) | 4 (2.5) | 1 (8.3) | – | – |
| Limb | 61 (59.8) | 114 (70.8) | 3 (25) | 13 (59.1) | 11 (84.6) |
| Limb and Other | 10 (9.8) | 12 (7.5) | – | 1 (4.5) | 1 (7.7) |
| Other | 1 (.88) | 6 (3.7) | 4 (33.3) | – | 1 (7.7) |
| College Education or greater, N (%) | 91 (80.5) | 105 (63.3) | 9 (69.2) | 20 (90.9) | 10 (76.9) |
| Mutation Carrier, N (%) | | | | | |
| C9ORF72 | 5 (4.4) | 17 (10.2) | 3 (20.0) | 0 (0.0) | 0 (0.0) |
| C9ORF72 and UBQLN2 | 0 (0.0) | 1 (0.6) | 3 (20.0) | – | – |
| SOD1 | 4 (3.5) | 4 (2.4) | – | – | – |
| SQSTM1 | 1 (0.9) | 0 (0.0) | – | – | – |
| TARDBP | 0 (0.0) | 1 (0.6) | – | – | – |
| TBK1 | 0 (0.0) | 1 (0.6) | – | – | – |
| Baseline ALSFRS-R (0–48), M (SD) | 35.98 (6.46) | 34.32 (7.44) | 35.00 (5.99) | 36.50 (5.95) | 33.62 (7.83) |
| UMN Score (0–10), M (SD) | 2.84 (1.68) | 2.61 (1.67) | 2.45 (2.00) | 4.54 (1.33) | 0.87 (0.73) |
| LMN Score (0–10), M (SD) | 2.29 (1.42) | 2.71 (1.50) | 2.81 (1.76) | 0.59 (0.96) | 4.84 (1.93) |
| ECAS, M (SD) | | | | | |
| ALS-Specific (0–100) | 88.91 (4.48) | 75.51 (10.55) | 52.62 (12.07) | 87.95 (7.47) | 81.62 (11.61) |
| Language (0–28) | 27.55 (0.50) | 24.69 (2.90) | 21.38 (3.93) | 26.82 (1.97) | 26.62 (1.26) |
| Verbal Fluency (0–24) | 19.63 (2.45) | 14.57 (5.43) | 7.83 (5.36) | 26.82 (1.97) | 16.77 (4.36) |
| Executive (0–48) | 41.73 (3.07) | 36.25 (6.39) | 24.00 (10.51) | 26.82 (1.97) | 38.23 (7.50) |
| ALS-Non-Specific (0–36) | 29.29 (3.11) | 27.19 (3.97) | 19.69 (8.30) | 29.73 (2.76) | 27.62 (6.31) |
| Memory (0–24) | 17.53 (3.08) | 15.71 (3.66) | 9.46 (7.15) | 17.95 (2.84) | 15.69 (6.20) |
| Visuospatial (0–12) | 11.76 (0.49) | 11.48 (0.92) | 11.08 (1.24) | 11.77 (0.43) | 11.92 (0.28) |
| Total (0–136) | 118.29 (6.32) | 102.69 (12.66) | 72.31 (18.53) | 117.68 (9.12) | 109.23 (16.47) |

PGB, Phenotype–Genotype–Biomarker; CReATe, Clinical Research in ALS and Related Disorders for Therapeutic Development; ALS, amyotrophic lateral sclerosis; ALSci, ALS-cognitive impairment; ALS-FTD, ALS frontotemporal dementia; PLS, primary lateral sclerosis, PMA, progressive muscular atrophy; ALSFRS-R, Revised ALS Functional Rating Scale; UMN, upper motor neuron; LMN, lower motor neuron; ECAS, Edinburgh Cognitive and Behavioral ALS Screen; M, mean, SD, standard deviation.

(PMA)) participating in the PGB study of the CReATe Consortium (NCT02327845) (Table 1). We included a spectrum of ALS and related disorder cases in an effort to account for the possibility that a subset of PLS or PMA cases may evolve into ALS (Kim et al, 2009) and can have similar cognitive profiles of cognitive dysfunction to ALS (de Vries et al, 2019). We used linear mixed effects (LME) to model variability between individuals in baseline performance and rate of decline on the ECAS (Total, ALS-Specific, and ALS-Non-Specific scores, and scores for each individual cognitive domain), on the ALS Functional Rating Scale—Revised (ALSFRS-R), and on clinician ratings of upper motor neuron (UMN) and lower motor neuron (LMN) signs (UMN and LMN burden scores); each model included covariate adjustment for potential confounders including age, education, bulbar onset, and disease duration. We confirmed that cognitive and motor performance at baseline are heterogeneous across individuals (Fig 1A), and correlation analyses of both baseline and longitudinal rates of change suggest that heterogeneity in cognition is independent of disability in physical function or clinical burden of UMN/LMN signs (all $R < 0.2$; Fig 1B). Together this establishes the heterogeneity of baseline and longitudinal cognitive and motor phenotypes within the PGB cohort.

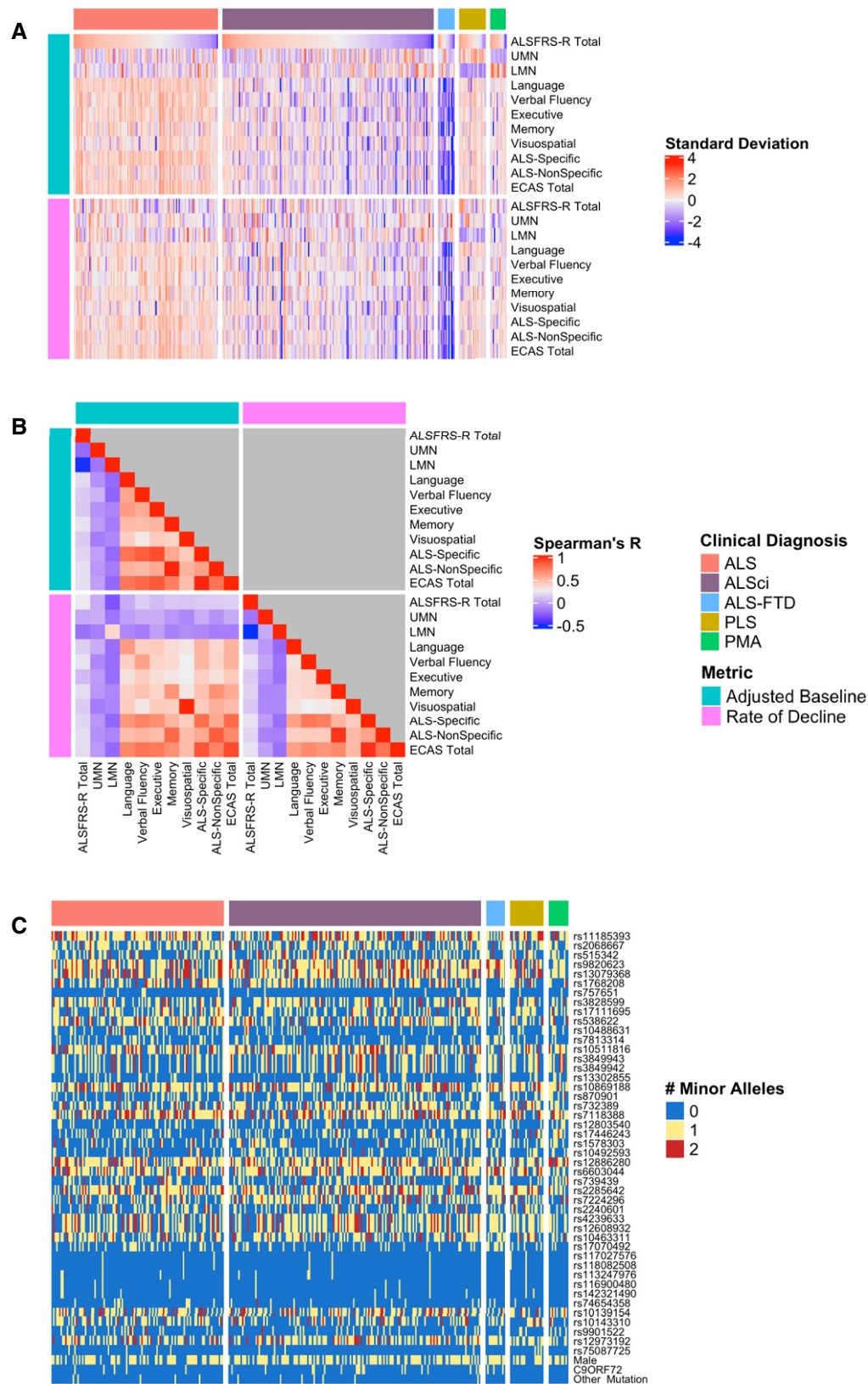

**Figure 1.**

**Figure 1.  Clinical and genetic heterogeneity in the CReATe PGB cohort.**

A  Differences in baseline performance and rate of decline on each clinical measure for each participant; the heatmap indicates each participant's standard deviation (SD) from the group mean.

B  Spearman's correlations between baseline performance and rate of decline for all clinical measures.

C  Allele dosage or binary status for each genetic variable for each participant.

## Multivariate analyses indicate polygenic contributions to baseline cognitive performance

To identify potential polygenic contributions to cognitive impairment in ALS, we employed sCCA (Witten *et al*, 2009), an unsupervised machine-learning approach enabling identification of multivariate relationships between a dataset of one modality (e.g., genetic variables including allele dosage of SNPs) and another modality (e.g., clinical measures of cognitive and motor function). Traditional CCA identifies a linear combination of all variables that maximize the correlation between datasets, resulting in an association of variables from one dataset (e.g., SNPs) and variables from another dataset (e.g., clinical scores) (Witten *et al*, 2009). The "sparse" component of sCCA additionally incorporates an L1 penalty that shrinks the absolute value of the magnitude of coefficients to yield sparse models (i.e., models with fewer variables) such that some coefficients are zero, and the variables associated with them are effectively eliminated from the model. As a result, variables that contribute little variance to the model are dropped, resulting in the identification of a data-driven subset of variables from one dataset that relate to a data-driven subset of variables from another dataset. Unstandardized regression coefficients resulting from sCCA serve as canonical weights indicating the direction and strength of the relationships between selected variables.

We evaluated an allele dosage dataset comprised of 33 SNPs identified as shared risk loci for both ALS and FTD (Karch *et al*, 2018), and 12 SNPs identified as risk loci for ALS from the largest published case–control GWAS (Nicolas *et al*, 2018), with the latter chosen to include loci associated with ALS but not specifically with FTD (Fig 1C). We included the first two principal components from a PCA conducted in the PGB cohort (Appendix Fig S1) and binary variables for sex, *C9ORF72* repeat expansion status, and other mutation status (e.g., *SOD1*) in this dataset in an effort to account for inter-individual genetic differences in population structure, sex, and mutation status. We then used sCCA to examine the association between this genetic dataset and a dataset comprised of adjusted baseline performance on clinical measures of cognitive and motor performance extracted from the LME models.

After optimizing model sparsity parameters (Fig EV1), we ran sCCA 10,000 times and employed random bootstrapped subsamples of 75% of participants in each iteration (Fig EV2). We then calculated the median canonical correlation between the clinical and genetic datasets, the median canonical weight for each variable in the genetic dataset, and the proportion of times (as a percentage) each variable from the clinical dataset was chosen out of 10,000 iterations. We report percentages rather than median canonical weight for clinical features because the optimized L1 parameter for the clinical dataset was the most stringent (i.e., 0.1), thus resulting in only one variable from the clinical dataset being chosen in each of the

10,000 iterations. This differs from other regularization techniques (e.g., LASSO), as the variable from the clinical dataset was selected by sCCA modeling in each iteration rather than being experimenter-selected prior to analysis. Importantly, the use of sCCA also minimizes the necessity for multiple comparison corrections, since all variables can be tested in a single model, and therefore reduces the potential of a type II false-negative error common in genomics studies related to rejection of a true effect due to overly stringent correction of multiple comparisons.

To assess model performance under the null hypothesis (no association between genetic factors and clinical phenotypes), we similarly ran 10,000 bootstrapped sCCAs using the same L1 and subsampling parameters and randomly permuted each dataset 100 times in each model iteration. We examined the proportion of times each variable in the clinical and genetic datasets was selected by this null model (i.e., achieving a non-zero canonical weight). We used the null model to define a $P$ value for the true, unpermuted model by calculating the probability under the null hypothesis of observing a canonical correlation greater than or equal to the median canonical correlation under sCCA modeling of the true data.

We observed that a subset of 29 genetic variables were correlated with a single clinical variable, achieving a median canonical correlation between the two datasets of $R = 0.35$ (95% confidence interval: 0.23, 0.42; $P = 0.019$) (Fig 2, Appendix Fig S2). Over the 10,000 iterations, the most frequently selected clinical variable was the ECAS ALS-Specific score (percentage of times selected: 37%), followed by the ECAS Total (29%), Executive Function (17%), Language (9.5%), Verbal Fluency (2.3%), ALS-Specific (2.2%), Memory (2%), and Visuospatial (0.34%) scores. The ALSFRS-R and UMN and LMN burden scores were each selected in < 0.05% of the model iterations. By contrast, performance of sCCA modeling under the null hypothesis demonstrated that each clinical variable was selected in a largely equal percentage of iterations (all variables ranging 5.9–9.4%), demonstrating that the true sCCA modeling selected cognitive and not motor features beyond what would be expected by chance (Fig EV3A).

Of the 29 selected genetic variables, the 12 most highly weighted were rs1768208 and rs9820623 (*MOBP*), rs7224296 (*NSF*), rs538622 (*ERGIC1*), rs10143310 (*ATXN3*), rs6603044 (*BTBD1*), rs4239633 (*UNC13A*), rs2068667 (*NFASC*), rs10488631 (*TNPO3*), rs11185393 (*AMY1A*), rs3828599 (*GPX3*), and sex. Twenty-seven of the 29 genetic variables selected were SNPs, and 85% of model-selected SNPs (23/27) were shared risk loci for ALS and FTD (Karch *et al*, 2018). Modeling under the null hypothesis revealed that each genetic variable achieved a largely equal median weight, and thus, there were no stronger model contributions from any subset of genetic variables (Fig EV3B). The association of genetic variables most frequently with the ECAS ALS-Specific score suggests polygenic contribution to impairment in domains of cognition frequently impaired in patients with ALS (e.g., language, verbal fluency, and

executive function), that are also the most impaired domains of cognition observed in FTD.

To evaluate whether our observed sCCA model was impacted by inclusion of patients with disorders related to ALS (i.e., PLS, PMA), we compared the median weights for genetic features and the percentage of times selected for clinical features from sCCA modeling using the entire CReATe PGB cohort (i.e., with PLS and PMA included) to those obtained from sCCA modeling using a subset of the CReATe PGB cohort that excluded patients with PLS and PMA. sCCA modeling that excluded patients with PLS and PMA resulted in the most frequent selection of the ECAS Total, ALS-Specific, Executive Function, and Language scores, similar to results obtained in the entire cohort (Appendix Fig S3A). Furthermore, sCCA modeling that excluded patients with PLS and PMA resulted in the same selection of genetic variables as in sCCA modeling of the entire cohort

and achieved similar direction and strength of weights (Appendix Fig S3B). This demonstrates that the inclusion of disorders related to ALS does not potentially confound our observations.

**Polygenic score captures baseline cognition as well as longitudinal rate of cognitive decline, but not motor decline**

Next, we investigated potential polygenic contributions to rate of decline in cognitive and motor performance in the PGB cohort. Investigation of baseline performance may only capture differences at a single (somewhat arbitrary) point in time, but not differences in the trajectory of performance over time.

To evaluate association with longitudinal performance, we first calculated a weighted polygenic score (wPRS) by computing a sum of allele dosage for each individual genetic variable multiplied by

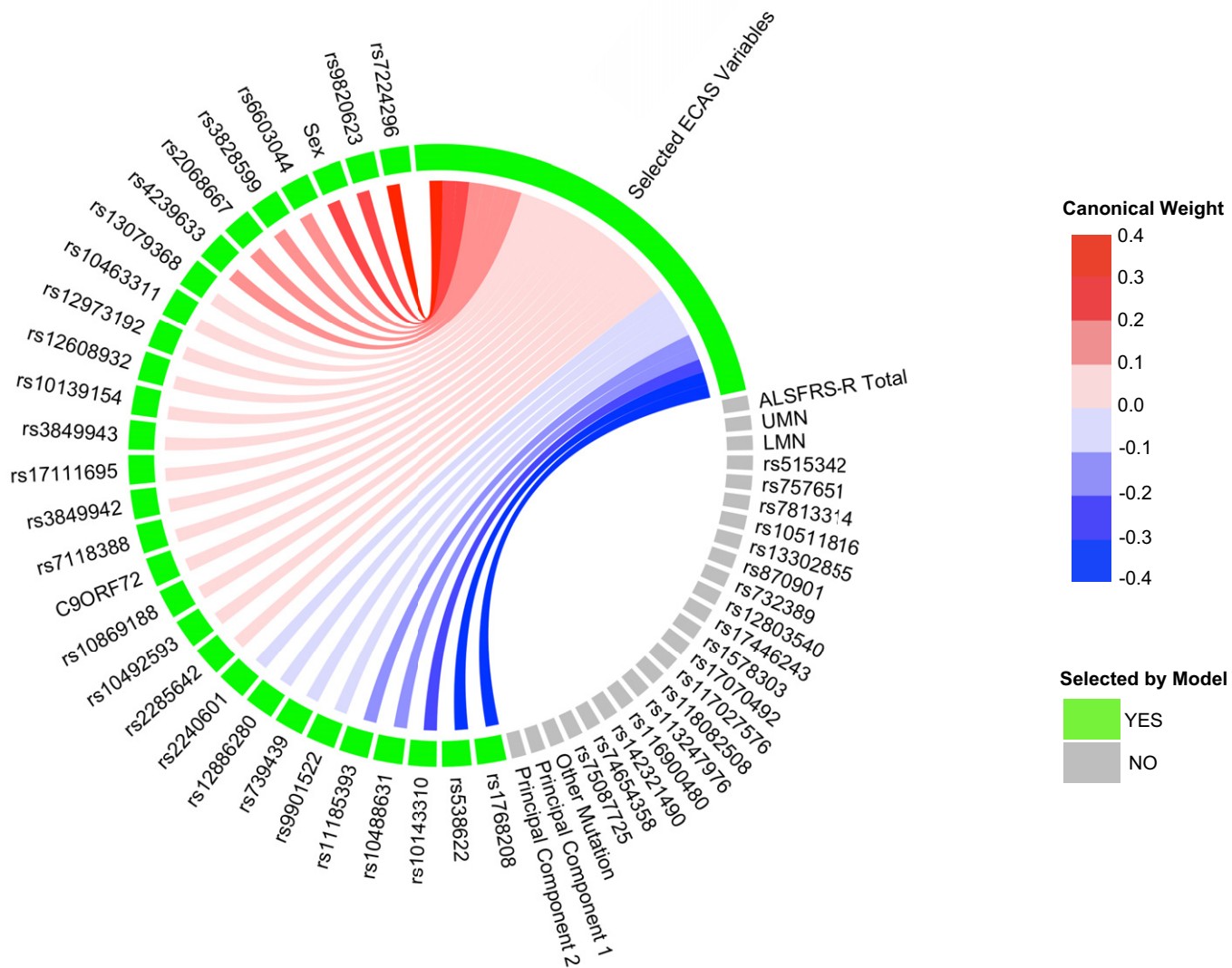

**Figure 2. Sparse, polygenic relationship between clinical and genetic variation in ALS.**
Variable selection and median canonical weight strength from bootstrap sCCA modeling in the CReATe PGB cohort. See Table EV1 for additional detail on genetic variants.

their median canonical weights from sCCA modeling. We also calculated an unweighted polygenic risk score (uPRS) by computing a sum of allele dosage for each individual genetic variable selected from sCCA modeling. Spearman rank-order correlations between the wPRS and adjusted baseline estimates of the four clinical features selected in 10% or more of the 10,000 iterations (e.g., ALS-Specific, Total, Executive Function, and Language scores from the ECAS) using family-wise error (FWE) correction resulted in correlation values similar to the median canonical correlation observed from sCCA modeling (e.g., for ECAS ALS-Specific: $rs(329) = -0.34$, $P = 5.0 \times 10^{-9}$) (Fig 3A), suggesting construct validity. We observed no statistically significant relationship between the uPRS and adjusted baseline estimates of performance on ALS-Specific, Executive Function, Language, and ECAS Total scores (all $P$ values > .2).

We then conducted Spearman's rank-order correlations between the wPRS and adjusted rate of decline on each clinical measure of cognitive and motor performance using FWE correction. To obtain adjusted rates of decline, we extracted individual slope estimates from prior LME (see above) for the 277 individuals (85%) from the PGB cohort with 2 or more observations on the ECAS, ALSFRS-R, and UMN and LMN burden scores. We observed significant negative relationships between the wPRS and adjusted rate of decline on ECAS ALS-Specific ($rs(277) = -0.21$, $P = 5.3 \times 10^{-3}$), ALS-Non-Specific ($rs(277) = -0.19$, $P = 0.016$), and Total scores ($rs(277) = -0.26$, $P = 8.1 \times 10^{-5}$; Fig 3B), but not on the ALSFRS-R or UMN and LMN burden scores (all $P > 0.9$). We observed no statistically significant relationship between the uPRS and adjusted rate of decline on any clinical measure (all $P$ values > .9). These findings suggest polygenic contribution using sCCA-derived weights to the rate of cognitive—but not motor—decline from the SNPs associated with risk of ALS or joint risk of ALS and FTD that were included in this analysis.

In *post hoc* analyses, we investigated whether SNPs also contribute individually to rate of decline on clinical measures (Fig EV4). We conducted LME modeling of the original longitudinal data to investigate fixed effects of each of the 45 SNPs on each of the 11 clinical measures (i.e., all ECAS scores, ALSFRS-R, and UMN and LMN burden scores), independently. We did not observe any effects that survived corrections for multiple comparisons. However, we observed that the SNPs achieving the five largest median weights from bootstrapped sCCA modeling (rs1768208, rs538622, rs10143310, rs7224296, and rs9820623) also independently related to performance on the ECAS ALS-Specific and Total scores (all uncorrected $P < 0.05$).

We also conducted *post hoc* analyses to investigate whether the inclusion of SNPs in high linkage disequilibrium (LD) influence the magnitude and direction of the wPRS we re-ran bootstrapped sCCA analyses using 10,000 iterations excluding the 5 SNPs in high LD (i.e., based on the cutoff of $R^2 > 0.5$) and recalculated the wPRS in the PGB cohort. This revealed a strong linear relationship between both wPRS models (Pearson's $R = 0.90$ (95% CI: 0.87, 0.91), $P < 2.2 \times 10^{-16}$; Appendix Fig S4), and thus, LD of a subset of SNPs is unlikely to be a driver of our observed polygenic associations.

## Polygenic score associates with cortical thinning in the UPenn Biobank

Cognitive dysfunction in ALS, including performance on the ECAS, has previously been attributed to sequential disease progression

rostrally and caudally from the motor cortex (Lulé *et al*, 2018) and to advancing disease stage (Crockford *et al*, 2018). To evaluate the neuroanatomical basis for polygenic contribution to cognitive performance in patients with ALS, we applied the wPRS score derived in the CReATe PGB cohort to an independent cohort of patients with ALS from the UPenn Biobank. We used voxel-wise *in vivo* measures of reduced cortical thickness (in mm$^3$) to quantify cortical neurodegeneration. Cross-sectional measurements of cortical thickness were derived from T1-weighted magnetic resonance imaging (MRI) in 90 patients with ALS and 90 age, sex, and education-matched healthy controls who were recruited for research from UPenn (Table 2). Nonparametric modeling using 10,000 random permutations revealed extensive reduction of cortical thickness bilaterally in the frontal and temporal cortices of patients relative to controls (threshold-free cluster enhancement, FWE corrected $P < 0.05$) (Table EV3, Appendix Fig S5).

After identifying regions of reduced cortical thickness in patients with ALS, we investigated whether the wPRS derived from sCCA modeling in the CReATe PGB cohort contributed to the magnitude of reduced cortical thickness in the independent UPenn Biobank neuroimaging cohort. Nonparametric modeling using 10,000 random permutations with adjustments for potential confounds in age, disease duration, and scanning acquisition revealed that a higher wPRS (i.e., greater risk) associated with greater reduction of cortical thickness in regions including the orbital prefrontal cortex, anterior cingulate cortex, premotor cortex, lateral temporal cortex, and hippocampus that survived uncorrected $P$ value of 0.01 and a cluster extent threshold of 10 voxels (Fig 4A; Table EV3). The frontal and temporal lobe cortical regions identified in this analysis are known to support the domains of cognitive dysfunction characterized by the ECAS (Lulé *et al*, 2018). We observed no statistically significant relationship between the uPRS and cortical thickness in any region. These findings provide a potential neuroanatomical basis for the observed polygenic relationships between the wPRS and baseline cognitive performance and rate of decline and are consistent with prior associations of cortical neurodegeneration with cognitive dysfunction in patients with ALS (Agosta *et al*, 2016).

## Polygenic score associates with neocortical neuronal loss in the UPenn Biobank

To complement these *in vivo* neuroanatomical data, we also explored whether polygenic risk for cognitive dysfunction associated with *post-mortem* anatomical distribution of neuronal loss and TDP-43 pathology. We assessed the magnitude of neuronal loss and TDP-43 pathological inclusions on an ordinal scale in tissue sampled from the middle frontal, cingulate, motor, and superior/middle temporal cortices and from the cornu ammonis 1 (CA1)/ subiculum of the hippocampus in 87 autopsy cases from the UPenn Biobank with confirmed ALS due to underlying TDP-43 pathology (Table 2; Table EV4). We conducted ordinal logistic regression with covariate adjustment for age at death and disease duration and found that as patients' wPRS increases, their odds of greater neuronal loss in the motor cortex also increases (OR = 1.98; 95% CI: 1.01, 3.96; uncorrected $P = 0.049$; Fig 4B); older age at death and longer disease duration were not found to statistically significantly influence these odds (uncorrected $P > 0.05$). We observed no statistically significant associations

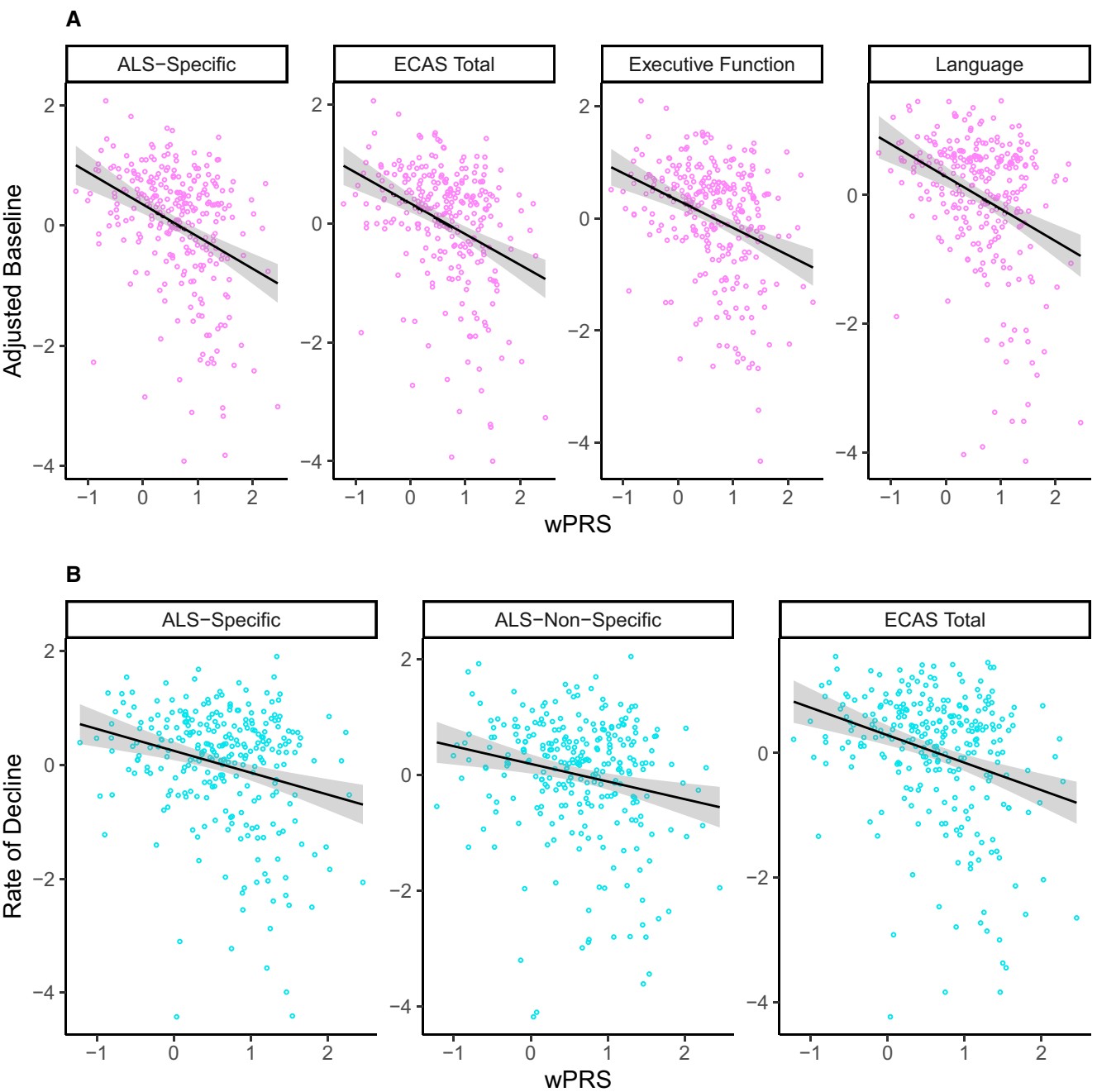

**Figure 3. wPRS correlates with cognitive performance on the ECAS in the CReATe PGB cohort.**

A, B Scatterplots showing that the wPRS correlates with (A) adjusted baseline performance on the ECAS ALS-Specific, Total, Executive Function, and Language scores, and (B) rate of decline on the ALS-Specific, ALS-Non-Specific, and Total scores.

between the wPRS and neuronal loss in any other region, or between the wPRS and TDP-43 pathology in any region (all *P* values > 0.1; Fig EV5 and Appendix Fig S6). We also observed no statistically significant associations between the uPRS and neuronal loss or TDP-43 pathology in region (all *P* values > 0.19, uncorrected). These findings suggest that polygenic risk for cognitive dysfunction is associated with the neuroanatomical distribution of neuronal loss in ALS cases at end-stage disease.

# Discussion

In this study, we evaluated polygenic contributions to cognitive dysfunction in patients with ALS by employing machine learning. We identified polygenic risk for cognitive dysfunction from genetic variables associated with risk of ALS and FTD, which we further investigated through quantitative-trait evaluations of two independent ALS cohorts with *in vivo* neuroimaging and *post-mortem*

**Table 2.  Demographics for independent neuroimaging (A) and autopsy (B) amyotrophic lateral sclerosis (ALS) and healthy control cohorts from UPenn Biobank.**

| A. Neuroimaging Cohort | | |
| --- | --- | --- |
| | **ALS** | **Healthy Control** |
| *N* (Male) | 90 (50) | 90 (38) |
| Diagnosis, *N* (%) | | |
| ALS | 54 (60.0) | |
| ALSci | 14 (15.60) | |
| ALS-FTD | 22 (24.40) | |
| Age at MRI in Years, M (SD) | 58.60 (11.10) | 61.09 (11.77) |
| Education in Years, M (SD) | 15.22 (3.12) | 15.89 (2.55) |
| Disease Duration in Years, M (SD) | 2.49 (1.88) | – |
| Mutation Carrier, *N* (%) | | |
| C9ORF72 | 13 (14.44) | – |
| SOD1 | 1 (1.11) | – |
| VCP | 1 (1.11) | – |
| Site of Symptom Onset, *N* (%) | | |
| Bulbar | 22 (24.44) | – |
| Limb | 60 (66.67) | – |
| Cognitive | 7 (7.78) | – |
| ALSFRS-R, M (SD) | 34.18 (7.46) | – |
| **B. Autopsy Cohort** | | |
| *N* (Male) | | 87 (48) |
| Diagnosis, *N* (%) | | |
| ALS | | 80 (91.95) |
| ALSci | | 5 (5.74) |
| ALS-FTD | | 2 (2.30) |
| Age at Death Years, M (SD) | | 63.83 (10.25) |
| Disease Duration at Death in Years, M (SD) | | 4.22 (3.43) |
| Mutation Carrier, *N* (%) | | |
| C9ORF72 | | 15 (17.24) |
| Site of Symptom Onset, *N* (%) | | |
| Bulbar | | 23 (26.44) |
| Limb | | 59 (67.82) |
| Cognitive | | 3 (3.45) |
| Respiratory | | 1 (1.15) |
| Unknown | | 1 (1.15) |

M, Mean; SD, standard deviation; ALSci, ALS-cognitive impairment; ALS-FTD, ALS with frontotemporal dementia.

neuropathology data. Our results indicate a polygenic contribution to the presence and rate of decline of cognitive dysfunction in domains specifically impaired in ALS. Converging evidence from these independent cohorts further demonstrates the generalizability of polygenic contribution to biologically plausible associations including reduced *in vivo* cortical thickness and *post-mortem* cortical neurodegeneration in the prefrontal, motor, and temporal cortices.

These findings contribute novel evidence in support of the polygenic contribution to cognitive dysfunction and cortical disease burden in ALS and provide further detailed phenotypic evidence for genetic overlap between ALS and FTD. Below, we highlight clinical, biological, and methodological implications for our observations.

Our findings add to an increasing body of evidence for a genetic contribution to phenotypic variability in ALS and support the idea that polygenic variation accounts for a portion of variability in cognitive dysfunction and cortical disease burden in ALS. While cognitive dysfunction has been more frequently linked to genetic mutations causally associated with ALS, such as *C9ORF72* repeat expansions (Byrne *et al*, 2012), studies examining individual SNPs have demonstrated quantitative-trait modification of cognitive performance and cortical disease burden (Vass *et al*, 2011; Placek *et al*, 2019). However, mounting evidence suggests that there are polygenic, rather than single allele, modifiers of disease risk and phenotype in ALS and related neurodegenerative diseases (Bandres-Ciga *et al*, 2019). Our observation of polygenic association between 27 SNPs and the ECAS ALS-Specific score, a combined measure of executive, language, and verbal fluency domains most commonly affected in ALS, is consistent with the idea of polygenic contribution to phenotypic variability in ALS. Notably, our observed polygenic association in the CReATe PGB cohort appears specific to cognitive variability: We demonstrate relative independence of cognitive performance and motor disease severity (i.e., UMN or LMN burden scores, functional performance on the ALSFRS-R) and observe no evidence for polygenic association with motor disease severity. This suggests that, in this study, polygenic risk for cognitive dysfunction does not appear to be confounded by motor disease severity.

The majority (85%) of the 27 SNPs selected by our machine-learning modeling for association with cognitive dysfunction are shared risk loci for ALS and FTD (Karch *et al*, 2018). The selection frequency of these ALS and FTD risk variants outweighed the selection of ALS-only risk variants, emphasizing the contribution of genetic overlap between ALS and FTD to polygenic risk associated with cognitive dysfunction in ALS. SNPs in or near the *MOBP, NSF, ATXN3, ERGIC1,* and *UNC13A* genes were among those with the strongest model contributions (i.e., with the largest canonical weights). Our group has previously shown that SNPs mapped to *MOBP*, including rs1768208, relate to regional neurodegeneration in sporadic FTD and to shorter survival in FTD with underlying tau or TDP-43 pathology (Irwin *et al*, 2014; McMillan *et al*, 2014). Our group has also demonstrated that rs12608932 in *UNC13A* relates to *in vivo* prefrontal cortical thinning, *post-mortem* frontal cortical burden of TDP-43 pathology, and executive dysfunction (Placek *et al*, 2019). rs538622 near *ERGIC1*, originally identified as a shared risk locus for ALS and FTD, has also previously been demonstrated to contribute to quantitative-trait modification in ALS by relating to reduced expression of the protein BNIP1 in ALS patient motor neurons (Karch *et al*, 2018). Other top-weighted variants near *NSF* and *ATXN3* indicate potential biological plausibility: rs10143310 is found near *ATXN3* which encodes a de-ubiquitinating enzyme, and polyglutamine expansions in *ATXN3* cause spinocerebellar ataxia—type 3 (Burnett *et al*, 2003); rs7224296 near *NSF* tags the *MAPT* H1 haplotype (Yokoyama *et al*, 2017) and is associated with increased risk for FTD syndromes including progressive supranuclear palsy and corticobasal degeneration (Ferrari et al, 2017), as well as Alzheimer's and Parkinson's diseases (Desikan *et al*, 2015).

**A**

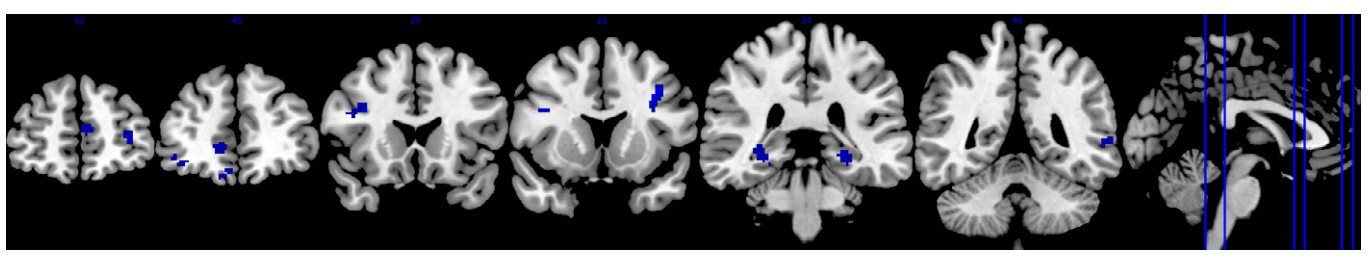

0.0                                          3.99

**B**

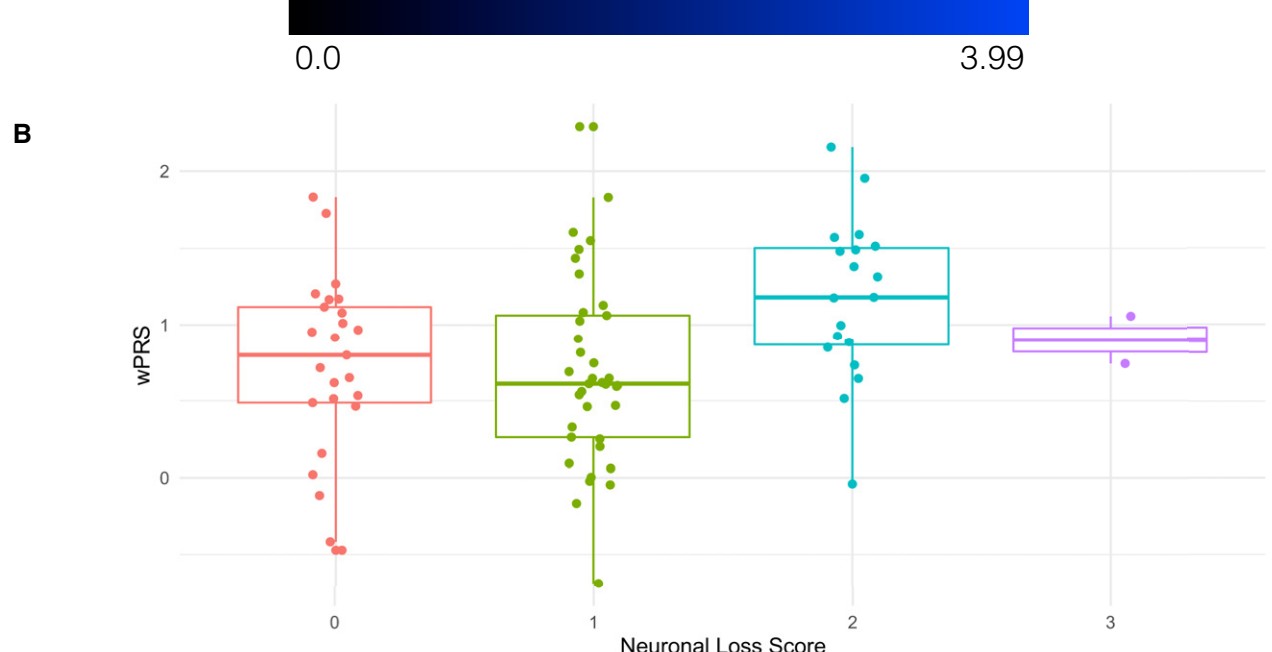

**Figure 4. Reduced cortical thickness and greater cortical neuronal loss relates to higher wPRS in independent validation cohorts.**

A   ALS patients from the UPenn Biobank neuroimaging cohort with higher wPRS exhibited greater reduction of cortical thickness in the orbital prefrontal cortex, anterior cingulate cortex, premotor cortex, lateral temporal cortex, and hippocampus. The heatmap indicates the associated T-statistic for each voxel, with light blue representing the highest value.

B   Magnitude of motor cortex neuronal loss in ALS cases from the UPenn Biobank is associated with higher wPRS. The central bands indicate the median, the box indicates the interquartile range (IQR) between the first and third quartiles, and the upper and lower whiskers indicate the largest or smallest value no further than the IQR multiplied by 1.5. Each data point in the boxplot represents a unique case.

While the mechanism of polygenic contribution to cognitive dysfunction in ALS requires further investigation, we speculate based on our findings that identified SNPs may contribute to neuroanatomical disease burden. The wPRS derived from the observed multivariate genotype–phenotype correlation in the CReATe PGB cohort showed robust relationships in independent cohorts from the UPenn Biobank to both *in vivo* cortical thinning and *post-mortem* cortical neuronal loss. Higher polygenic risk related to *in vivo* cortical thinning in the orbital prefrontal cortex, anterior cingulate cortex, premotor cortex, lateral temporal cortex, and hippocampus in a neuroimaging cohort, and to *post-mortem* neuronal loss in sampled tissue from the motor cortex in an autopsy cohort. We speculate that the relationship to motor cortex only in the neuropathology cohort may reflect two sources of sampling differences. First, clinical characteristics differed across cohorts: 9% of

the autopsy cohort had premorbid diagnoses of ALS-FTD or ALSci and 29% of the neuroimaging cohort were diagnosed with ALS-FTD or ALSci. Thus, the autopsy cohort likely had less frontal and temporal cortex neuronal loss relative to motor cortex neuronal loss. Second, the differences across analyses may reflect different scales of resolution in which neuroimaging data are analyzed at 2 mm³ resolution across the entire cortex while neuropathological data are sampled at approximately 6 μm. We are aware of these issues and more recently have begun to increase tissue sampling including bilateral hemisphere (Irwin *et al*, 2018; Giannini *et al*, 2019), more extensive brain regions (Irwin *et al*, 2016a), performing digital immunohistochemistry analyses (Irwin *et al*, 2016b; Giannini *et al*, 2019), and whole hemisphere post-mortem neuroimaging using 7T MRI. Thus, future studies will be able to address these sampling differences as our autopsy cohort continues to grow and our technical

methods continue to improve. Anatomically, these findings are largely consistent with prior *in vivo* structural imaging studies of neurodegeneration associated with cognitive dysfunction and with *post-mortem* investigations of cortical thinning in ALS (Prudlo *et al*, 2016; Lulé *et al*, 2018). Thus, in addition to indicating polygenic contribution to cognitive dysfunction in ALS, our findings suggest a possible mechanism of observed findings via disease pathophysiology.

Beyond the potential biological mechanism of identifying polygenic contributions to ALS disease heterogeneity, we additionally suggest that sCCA may provide a tool for defining polygenic factors of disease risk. While sCCA has been widely applied to genotype–phenotype studies (Witten & Tibshirani, 2009), including neuroimaging-genetic studies (McMillan *et al*, 2014; Hao *et al*, 2017), we are unaware of prior applications using sCCA to define a polygenic score based on rich clinical phenotypic and biomarker data. Traditional approaches to the generation of polygenic scores include using data from established, typically case–control GWAS, but practical considerations involve the selection of how many variants to include in a model and how to define the weights of an appropriate statistical model (Sugrue & Desikan, 2019). Critically, rather than an arbitrary selection of variants and their weights, the sparsity parameter of sCCA facilitates an unsupervised, data-driven method to select the number of variants to include and also provides data-driven canonical weights to define the statistical model. The positive or negative direction of model-derived weights is potentially biologically informative, and could reflect "risk" (i.e., positive weight) or "protective" (i.e., a negative weight) effects. We evaluated the wPRS using model-derived weights relative to a uPRS-derived created by computing an unweighted sum of allele dosages for each genetic variable. Our observation that the uPRS did not relate to cognitive or clinical performance in the CReATe PGB cohort or to neuroimaging or neuropathology in the UPenn Biomarker cohorts suggests that that the weights derived from sCCA meaningfully define the relationship between genetic variation and quantitative phenotypic differences in the CReATe PGB and UPenn cohorts with regard to cognitive performance and disease neuroanatomy. Further investigation is needed to clarify the relationships between model-selected SNPs and model-derived canonical weights from both biological (e.g., some SNPs and/or genes may contribute more strongly to risk factors) and mathematical (e.g., weights may be constrained by minor allele frequency) perspectives. While our sCCA modeling selected 27 SNPs in addition to sex and *C9ORF72* mutation status and we used model-derived weights to calculate a wPRS, we are unable to determine in the current study what the collective contribution of these SNPs are to modifying cognitive phenotypes. For example, these could be additive in nature, such that increased risk allele dosage increases risk for impaired cognition, or the selected SNPs could act independently in disease modification. *Post hoc* investigation of independent SNP effects on longitudinal cognitive performance revealed that the SNPs achieving the five largest median weights from bootstrapped sCCA modeling also relate to longitudinal cognitive performance; however, these effects did not survive correction for multiple comparisons. By its nature, this *post hoc* investigation considered each SNP as independent from other SNPs and each clinical measure as independent from other clinical measures and thus did not account for more complex collective contribution of SNPs to cognitive phenotypes. As

often is the case, future functional studies are required to identify the mechanistic relationship between SNP associations and cognitive phenotype. Nonetheless, our results support the consideration of sCCA as a promising method to identify collective combinations of SNPs and cognitive phenotypes and to direct research efforts toward model-selected variants.

Several limitations should be considered in the present study. Here, we focus our analysis on a relatively small set of SNPs selected *a priori* from previous large-scale GWAS based on genome-wide association with ALS (Nicolas *et al*, 2018) or shared risk between ALS and FTD (Karch *et al*, 2018). Other genetic variants not included in the present study may also contribute to cognitive dysfunction in ALS and related disorders, and future genome-wide analyses or broad genotype selection strategies (e.g., targeted pathways) are necessary to elucidate discovery of novel genetic contributions to cognition that have not been identified through prior case–control studies. While we focus on ALS and FTD risk variants and demonstrate that the inclusion of related disorders (i.e., PLS, PMA) does not confound our observed cognitive and genetic associations, future work should also incorporate variants associated with risk for disorders related to ALS and specifically test the application of polygenic associations within PLS and PMA. However, such larger scale studies will require validation in independent cohorts, many of which are lacking the rich phenotype data needed to identify cognitive dysfunction. We derived a wPRS from sCCA modeling to further investigate polygenic associations with longitudinal cognitive and motor performance, and with *in vivo* and *post-mortem* cortical disease burden in independent ALS cohorts from the UPenn Biobank. While we define our polygenic score from sCCA using adjusted estimates of baseline cognitive and motor performance, future work using longitudinal data as the starting point to define polygenic associations may further elucidate genetic risk for cognitive dysfunction in ALS. However, our finding that polygenic risk associated with baseline cognitive dysfunction also relates to longitudinal cognitive decline in the CReATe PGB cohort as well as to relevant cortical disease anatomy in independent cohorts from the UPenn Biobank suggests its relevance to longitudinal cognitive phenotypes in ALS. Previous critique of polygenic scores argue that three factors limit their use in clinical and prognostic settings: (i) calculation based on GWAS-defined odds ratios for univariate risk loci; (ii) undue influence by population variance; and (iii) predominant use of samples of European ancestry (Wald & Old, 2019; Duncan *et al*, 2019). In an attempt to mitigate these potential confounds, we based our computation of a wPRS on model-selected parameters derived from an analysis including all genetic variants and, in addition, covariates for genetic mutation status and sex in an effort to account for multivariate genetic relationships. We also included the first two principal components in our model from a PCA conducted in the CReATe PGB cohort in an effort to account for differences in population substructure (Price *et al*, 2006). While we used the first two principal components in an effort to account for population substructure, this is a complex issue to resolve and future studies with more diverse cohorts to investigate potential substructure bias are necessary. The current investigation utilized existing data from natural history studies that were predominantly comprised of individuals of European ancestry; however, increased representation of diverse racial and ethnic groups in future

investigations of polygenic risk for cognitive impairment in ALS is necessary in order to ensure generalizability to diverse populations.

Our analyses focused on the investigation of genetic contribution to cognitive dysfunction in ALS, yet it is well established that behavioral impairment is also part of the ALS spectrum disease (Lillo *et al*, 2010). We assessed patient performance on specific domains of cognition using the ECAS, which includes a measure of social cognition counted toward the domain of executive function. Behavioral impairment on the ECAS is assessed through caregiver report (Abrahams *et al*, 2014), and the vast majority of neuropsychological assessments of behavior in neurodegenerative disease are based on physician or caregiver report (Simon & Goldstein, 2019). With this in mind, we chose to focus our investigation on the analysis of patient-completed assessments of cognition and motor function. Future research incorporating assessments of behavior is necessary to investigate polygenic risk for behavioral dysfunction in ALS and related disorders and to determine whether loci included in our calculated polygenic risk score additionally confer risk for behavioral dysfunction. Although the current study demonstrates converging, multimodal evidence for polygenic risk, replication in additional cohorts with larger sample sizes that allow for robust cross-validation is warranted. Notably, machine-learning methods have the tendency to over-fit data and produce estimates that do not generalize to different datasets. However, alternative datasets for ALS that contain detailed genotyping and cognitive phenotyping are currently lacking and the CReATe PGB cohort represents the largest of its kind. In the absence of an alternative dataset to minimize over-fitting, we employed a bootstrapping procedure and generated a final sCCA model based on median weights across permutations rather than selecting a single "top model". We additionally demonstrate converging, multimodal evidence for polygenic risk in independent neuroimaging and neuropathology biomarker cohorts in an effort to provide corroboration that we are detecting a true biological signal. However, future research is necessary to determine the predictive potential and generalizability of our proposed polygenic risk score in ALS patients. We furthermore hope that this demonstration motivates the collection of additional genotyping data and longitudinal cognitive evaluation using the ECAS in additional large-scale patient cohorts.

With these limitations in mind, our research demonstrates converging clinical, neuroimaging, and pathologic evidence for polygenic contribution to cognitive dysfunction and cortical neurodegeneration in ALS. These findings should stimulate further investigation into polygenic risk for cognitive disease vulnerability in ALS and suggest their importance in prognostic consideration and treatment trials. More broadly, this work provides insight into genetic contribution to heterogeneous phenotypes in neurodegenerative disease and supports evidence for polygenic architecture in these conditions.

# Materials and Methods

## Participants: CReATe consortium

Participants consisted of 339 individuals clinically diagnosed by a board-certified neurologist with a sporadic or familial form of amyotrophic lateral sclerosis (ALS), amyotrophic lateral sclerosis with frontotemporal dementia (ALS-FTD), progressive muscular atrophy (PMA), or primary lateral sclerosis (PLS) who were enrolled and evaluated through the CReATe Consortium's Phenotype–Genotype–Biomarker (PGB) study. All participants provided written informed consent. The PGB study is registered on clinicaltrials.gov (NCT02327845) and the University of Miami Institutional Review Board (IRB) (the central IRB for the CReATe Consortium) approved the study. This study entails participant blood DNA samples available for genetic screening and longitudinal evaluation at regularly scheduled visits (ALS, ALS-FTD, and PMA: 0 (baseline), 3, 6, 12, and 18 months; PLS: 0 (baseline), 6, 12, 18, and 24 months). A subset of 155 CReATe PGB cases were previously included in the replication cohort of the ALS case–control GWAS (Nicolas *et al*, 2018). Participants were evaluated at each visit using the ALSFRS-R (Cedarbaum *et al*, 1999) and alternate versions of the Edinburgh Cognitive and Behavioural ALS Screen (ECAS) (Abrahams *et al*, 2014) designed for longitudinal use. Presence of ALS with cognitive impairment (ALSci) was assessed at baseline using the ECAS according to established criteria (Strong *et al*, 2017), operationalized as baseline performance on Executive Function, Verbal Fluency, or Language subscores at or below normative cutoff scores (Abrahams *et al*, 2014). UMN and LMN burden scores were calculated from a detailed elemental neuromuscular examination by summing within and across each spinal region resulting in a score ranging from 0 (none) to 10 (worst). Site (e.g., limb, bulbar) and date of motor symptom onset were recorded for each participant. We excluded nine individuals with missing or incomplete data that precluded subsequent analysis and, in an effort to avoid confounds associated with clear outliers, three individuals with extreme values at baseline on the ECAS Visuospatial Score (i.e., > 5 standard deviations from group mean), resulting in a total of 327 participants. Of the nine excluded individuals with missing or incomplete data, one had no genotyping data available, one had no information for UMN burden score, and seven had no information for date of motor symptom onset. All patients and controls participated in an informed consent procedure approved by an IRB convened at the University of Pennsylvania or University of Miami, in which all studies conformed to the principles set out in the WMA Declaration of Helsinki and the Department of Health and Human Services Belmont Report.

## Genotyping: CReATe consortium

Peripheral blood mononuclear cell DNA was extracted using the QIAamp DNA Blood Mini Kit Qiagen #51106 and quantified using the Quant-iT dsDNA Assay Kit (Life Technologies cat#Q33130). The DNA integrity was verified by agarose gel electrophoresis (E-Gel, Life Technologies, cat#G8008-01). Unique samples were barcoded and whole genome sequencing (WGS) was performed at the HudsonAlpha Institute for Biotechnology Genomic Services Laboratory (Huntsville, Alabama) (HA) using Illumina HiSeq X10 sequencers to generate approximately 360 million paired-end reads, each 150 base pairs (bp) in length. Peripheral DNA was extracted from participant blood samples and screened for known pathogenic mutations associated with ALS and related diseases.

Screening included repeat-primed polymerase chain reaction (PCR) for *C9ORF72* repeat expansions and WGS curated and validated via Sanger sequencing for pathogenic mutations associated

with ALS and/or FTD in *ANG, CHCHD10, CHMP2B, FUS, GRN, hnRNPA1, hnRNPA2B1, MAPT, MATR3, OPTN, PFN1, SETX, SOD1, SPG11, SQSTM1, TARDBP, TBK1, TUBA4A, UBQLN2, VCP* (see Table 1 for participant mutation status). The PGB study also includes patients with hereditary spastic paraplegia (HSP) that were excluded in the current analysis, but we additionally screened individuals for pathogenic mutations in 67 additional genes associated with HSP and 7 genes associated with distal hereditary motor neuropathy, and all cases were negative for pathogenic mutations in these genes.

Whole genome sequencing (WGS) data were generated using paired-end 150 bp reads aligned to the GRCh38 human reference using the Burrows–Wheeler Aligner (BWA-ALN v0.7.12) (Li & Durbin, 2010) and processed using the Genome Analysis Toolkit (GATK) best-practices workflow implemented in GATK v3.4.0 (McKenna *et al*, 2010). Variants for individual samples were called with HaplotypeCaller, producing individual variant call format files (gVCFs) that we combined using a joint genotyping step to produce a multi-sample VCF (pVCF). Variant filtration was performed using Variant Quality Score Recalibration (VQSR), which assigns a score to each variant and a pass/fail label and evaluated this in the context of hard filtering thresholds (minimum genotype quality (GQ) $\geq$ 20, minimum mean depth value (DP) $\geq$ 10). Variant annotation was performed using variant effect predictor (VEP) (Hunt *et al*, 2018) and in-house pipelines including non-coding variant allele frequencies from Genome Aggregation Database (gnomAD) (Karczewski *et al*, 2020). In-house scripts were used to identify false positives resulting from paralogous mapping or/and gaps in the current human genome assembly. VCFs were further decomposed prior to analyses using the Decompose function of Vt (Tan et al 2015). In an attempt to account for population substructure, we additionally derived the first two principal components scores for each in the CReATe PGB cohort using principal components analysis (PCA) implemented using Eigenstrat (Price *et al*, 2006).

From the WGS data, we extracted 45 hypothesized variants from WGS that previously achieved genome-wide significance for association with ALS (Nicolas *et al*, 2018) or joint association with ALS and FTD (Karch *et al*, 2018). Proxy loci were genotyped (LD $R^2 > 0.80$) when genetic data were not available for previously published loci (see Table EV1 for a complete list). One locus, rs12973192, was common to both references, and another locus (rs2425220 (Karch *et al*, 2018)) was excluded from analysis due to high level of missingness across samples; no LD proxy was identified. We then used PLINK software (Purcell *et al*, 2007) to recode participant genotypes according to additive genetic models (e.g., 0 = no minor allele copies, 1 = one minor allele copy, 2 = two minor allele copies), since the dominant or recessive nature of the loci included in this study remains unknown. An assessment of LD revealed that 5 of our 45 hypothesized SNPs were in high LD with one another (D'>0.8; Table EV2), but we included these high LD SNPs in our investigation since sCCA is able to accommodate highly correlated features (Witten *et al*, 2009).

## Linear mixed effects modeling of the ECAS and clinical measures

We conducted linear mixed effects modeling of performance on the ECAS, ALSFRS-R, and UMN and LMN burden scores using the *nlme* package in R. Each model was fit using maximum likelihood. In

addition to the ECAS Total Score, we analyzed Executive Function, Language, Verbal Fluency, Memory, and Visuospatial subscores and ALS-Specific and ALS-Non-Specific summary scores each as dependent variables to analyze patient performance in separate cognitive domains and in clinically grouped cognitive domains. Fixed effects included age at baseline visit (in years), lag between age of symptom onset and age at baseline visit (in years), college education (yes/ no), bulbar onset (yes/ no) and visit time-point (in months), and we included individual-by-visit time-point as a random effect. This allowed us to obtain adjusted estimates of baseline performance (i.e., intercept) and rate of decline (i.e., slope) per individual, having regressed out potential confounding variables as fixed effects.

We conducted Spearman's rank-order correlations between baseline performance and rate of decline using FWE correction for multiple comparisons (see Fig 1B).

In addition to the linear mixed effects models described above, we also conducted a second series of linear mixed effects models to investigate fixed effects of each of the 45 SNPs on each of the 11 clinical measures (i.e., all ECAS scores, ALSFRS-R, and UMN and LMN burden scores), independently; this resulted in a total of 495 models. We again used the *nlme* package in R and each model was fit using maximum likelihood. In addition to each SNP, we included age at baseline visit (in years), lag between age of symptom onset and age at baseline visit (in years), college education (yes/ no), bulbar onset (yes/ no) and visit time-point (in months) as fixed effects, and we included individual-by-visit time-point as a random effect.

## Sparse canonical correlation analysis

We conducted sparse canonical correlation analysis (sCCA) to select a parsimonious linear combination of variables that maximize the correlation between two multivariate datasets using the *PMA* package in R (Witten *et al*, 2009). The first dataset comprised scaled intercepts from each clinical variable per participant (i.e., adjusted baseline performance on the ALSFRS-R, UMN, and LMN assessments, and ECAS). The second comprised minor allele counts per individual for each of the 45 SNPs (e.g., 0 = no minor allele copies, 1 = one minor allele copy, 2 = two minor allele copies), binary variables for sex (0 = female, 1 = male), *C9ORF72* repeat expansion status (0 = noncarrier, 1 = carrier), and other mutation status (0 = noncarrier, 1 = carrier) and, in an effort to account for potential population differences in population substructure, we also included the raw estimates for the first two principal components per participant derived from a PCA conducted in the CReATe PGB cohort; this method has previously been demonstrated to account for the majority of population structure (Price *et al*, 2006).

We assumed standard (e.g., unordered) organization of each dataset and selected regularization parameters for the sCCA analysis using a grid search of 100 combinations of L1 values between 0 (most sparse) and 1 (least sparse) in increments of 0.1. We selected the combination of L1 values yielding the largest canonical correlation of the first variate for subsequent analysis, as similarly reported (Xia *et al*, 2018).

Using these L1 parameters, we ran 10,000 bootstrap sCCAs and in each iteration employed randomly generated subsamples comprising 75% of the PGB cohort. We calculated the median

canonical correlation for sCCA and the median canonical weights for each variable across all iterations. We utilized the median in these estimates rather than the maximum or mean value in an effort to avoid bias from outliers and to increase the reliability and reproducibility of model estimates.

We next investigated model performance under a null hypothesis (i.e., no association between clinical and genetic datasets) by using randomly permuted data. Using the same L1 parameters, we again ran 10,000 bootstrap sCCAs and in each iteration employed randomly generated subsamples of 75% of participants; however, in each iteration, we randomly permuted each dataset 100 times using the *randomizeMatrix* function from the *picante* package in R. We calculated a *P* value by reporting the probability under the null of observing a canonical correlation greater than or equal to the median canonical correlation under sCCA modeling of the true data. We also examined the proportion of iterations each variable was selected by the model (i.e., achieving a non-zero canonical weight).

## Polygenic risk score

We used the output of sCCA modeling to calculate a wPRS for each individual. A wPRS for each individual in the PGB cohort, and in the neuroimaging and autopsy UPenn Biobank cohorts, was constructed by multiplying allele dosage or binary coding at each genetic variable by its median canonical weight from sCCA modeling and summing across all values.

To investigate construct validity, we first conducted Spearman's rank-order correlations between the wPRS and adjusted estimates of baseline performance (i.e., LME-derived intercepts) on the most frequently selected clinical measure(s) selected from sCCA.

Then, to investigate longitudinal performance associated with the wPRS, we conducted Spearman's rank-order correlations between the wPRS and adjusted rates of decline (i.e., LME-derived slopes) on all clinical measures using FWE correction. We restricted this analysis to participants in the CReATe PGB cohort with data at 2 or more time-points (*N* = 277 out of 327 participants), or 84.7% of the cohort.

## Participants: UPenn Biobank neuroimaging cohort

We retrospectively evaluated 90 patients with ALS and 90 healthy controls matched for age, sex, and education from the UPenn Biobank who were recruited for research between 2006 and 2019 from the Penn Comprehensive ALS Clinic and Penn Frontotemporal Degeneration Center (Table 2) (Toledo *et al*, 2014). Inclusion criteria for ALS patients consisted of the following: lack of participation in the CReATe PGB cohort, complete genotyping at the 45 analyzed SNPs, screening for genetic mutations (e.g., *C9ORF72, SOD1*), white non-Latino racial and ethnic background (population diversity is known to influence allele frequencies across individuals), disease duration from symptom onset < 2.5 standard deviations from respective group means (to avoid confounds associated with clear outliers), and T1-weighted MRI. All patients were diagnosed with ALS by a board-certified neurologist (L.E., L.M., M.G., D.I.) using revised El Escorial criteria (Brooks *et al*, 2000) and assessed for ALS frontotemporal spectrum disorder using established criteria (Strong *et al*, 2017); those patients enrolled in research prior to 2017 were retrospectively evaluated through chart review. All ALS patients and

controls participated in an informed consent procedure approved by an IRB convened at UPenn, in which all studies conformed to the principles set out in the WMA Declaration of Helsinki and the Department of Health and Human Services Belmont Report.

## Participants: UPenn Biobank autopsy cohort

We evaluated brain tissue samples from 87 ALS autopsy cases identified from the UPenn Biobank (Toledo *et al*, 2014) who were diagnosed by a board-certified neuropathologist (J.Q.T., E.B.L.) with ALS due to TDP-43 pathology using immunohistochemistry (Neumann *et al*, 2006) and published criteria (Mackenzie *et al*, 2011); this cohort included 20 patients from the ALS neuroimaging cohort. During life, all patients were diagnosed with ALS by a board-certified neurologist (L.E., L.M., M.G., D.I.) using revised El Escorial criteria (Brooks *et al*, 2000) and assessed for ALS frontotemporal spectrum disorder using established criteria (Strong *et al*, 2017); those patients enrolled in research prior to 2017 were retrospectively evaluated through chart review. Inclusion criteria consisted of the following: lack of participation in the CReATe PGB cohort, complete genotyping at the 45 analyzed SNPs, screening for genetic mutations (e.g., *C9ORF72, SOD1*), white non-Latino racial and ethnic background (population diversity is known to influence allele frequencies across individuals), disease duration from symptom onset < 2.5 standard deviations from respective group means (to avoid confounds associated with clear outliers), and brain tissue samples from the middle frontal, motor, cingulate, and superior/temporal cortices, and the cornu ammonis 1 (CA1)/ subiculum of the hippocampus for analysis of neuronal loss and TDP-43 pathology. Nine individuals were missing neuronal loss or TDP-43 pathology data for at least one sampled region (Table EV3).

## Genetic screening and SNP genotyping: UPenn biobank

DNA was extracted from peripheral blood or frozen brain tissue following the manufacturer's protocols (Flexigene (Qiagen) or QuickGene DNA whole blood kit (Autogen) for blood, and QIAsymphony DNA Mini Kit (Qiagen) for brain tissue). All patients were screened for *C9ORF72* hexanucleotide repeat expansions using a modified repeat-primed PCR as previously described (Suh *et al*, 2015). Of the remaining individuals, we evaluated family history using a three-generation pedigree history, as previously reported (Wood *et al*, 2013). For cases with a family history of the same disease, we sequenced 45 genes previously associated with neurodegenerative disease, including genes known to be associated with ALS (e.g., *SOD1* (Rosen *et al*, 1993) and *TBK1* (Freischmidt *et al*, 2015). Sequencing was performed using a custom-targeted next-generation sequencing panel (MiND-Seq) (Toledo *et al*, 2014) and analyzed using Mutation Surveyor software (Soft Genetics, State College, PA).

DNA extracted from peripheral blood or cerebellar tissue samples was genotyped for each case using the Illumina Infinium Global Screening Array through the Children's Hospital of Philadelphia (CHOP) Center for Applied Genomics Core according to manufacturer's specifications. PLINK (Purcell *et al*, 2007) was then used to remove variants with < 95% call rate, Hardy–Weinberg equilibrium (HWE) *P* value < $10^{-6}$ and individuals with > 5% missing genotypes. Using the remaining genotypes from samples passing quality

control, we performed genome-wide imputation of allele dosages with the Haplotype Reference Consortium panel (r1.1) (McCarthy *et al*, 2016) on the Michigan Imputation Server (Das *et al*, 2016) to predict genotypes at ungenotyped genomic positions, applying strict pre-phasing, pre-imputation filtering, and variant position and strand alignment control.

**Neuroimaging processing and analyses**

High-resolution T1-weighted MPRAGE structural scans were acquired for neuroimaging participants using a 3T Siemens Tim Trio scanner with an 8-channel head coil, with T = 1,620 ms, T = 3.09 ms, flip angle = 15°, 192 × 256 matrix, and 1 mm$^3$ voxels. T1-weighted MRI images were then preprocessed using Advanced Normalization Tools (ANTs) software (Tustison *et al*, 2014). Each individual dataset was deformed into a standard local template space in a canonical stereotactic coordinate system. ANTs provide a highly accurate registration routine using symmetric and topology-preserving diffeomorphic deformations to minimize bias toward the reference space and to capture the deformation necessary to aggregate images in a common space. Then, we used N4 bias correction to minimize heterogeneity (Tustison *et al*, 2010) and the ANTs Atropos tool to segment images into six tissue classes (cortex, white matter, cerebrospinal fluid, subcortical gray structures, brainstem, and cerebellum) using template-based priors and to generate probability maps of each tissue. Voxel-wise cortical thickness was measured in millimeters (mm$^3$) from the pial surface and then transformed into Montreal Neurological Institute (MNI) space, smoothed using a three sigma full-width half-maximum Gaussian kernel, and downsampled to 2 mm isotropic voxels.

We used *randomise* software from FSL to perform nonparametric, permutation-based statistical analyses of cortical thickness images from the UPenn Biobank neuroimaging cohort. Permutation-based statistical testing is robust to concerns regarding multiple comparisons since, rather than a traditional assessment of two sample distributions, this method assesses a true assignment of factors (e.g., wPRS) to cortical thickness compared to many (e.g., 10,000) random assignments (Winkler *et al*, 2014).

First, we used *randomise* set to 10,000 permutations to identify reduced cortical thickness in ALS patients relative to healthy controls. We constrained this analysis using an explicit mask restricted to high probability cortex (> 0.4) and reported clusters that survive $P < 0.05$ threshold-free cluster enhancement (Smith & Nichols, 2009) corrected for FWE.

Next, we again used *randomise* set to 10,000 permutations to identify regions of reduced cortical thickness associated with wPRS in ALS patients, constraining analysis to an explicit mask defined by regions of reduced cortical thickness in ALS patients relative to controls (see above). The statistical model for this analysis included covariate adjustment for age, disease duration, and scanner acquisition. We report clusters that survive uncorrected $P < 0.01$ with a cluster extent threshold of 10 voxels; we employ an uncorrected threshold to minimize the chance of type II error (not observing a true result).

**Neuropathology processing and analyses**

The extent of neuronal loss and of phosphorylated TDP-43 intraneuronal inclusions (dots, wisps, skeins) in sampled regions from

**The paper explained**

**Problem**
Cognitive dysfunction is often observed in patients with amyotrophic lateral sclerosis (ALS), a progressive neuromuscular disease linked clinically, pathologically, and genetically to frontotemporal dementia (FTD). Many sources of common genetic variation associated with ALS and FTD have been identified, but polygenic contribution of these sources to cognitive dysfunction and corresponding cortical disease vulnerability in ALS has not yet been explored.

**Results**
We used a machine-learning approach to identify a subset of common genetic variants that relate to cognitive dysfunction in a large cohort of patients with ALS and related disorders. A polygenic risk score (PRS) created from the subset of common genetic variants using machine-learning results related to more severe brain atrophy in the frontal and temporal lobes in independent cohorts of ALS patients.

**Impact**
Polygenic architecture may underlie risk for cognitive disease vulnerability in ALS and related disorders, and machine-learning approaches provide a novel strategy for PRS generation. This is important not only for future research on genetic contribution to patient symptoms and disease progression, but also for prognostic consideration and clinical trial planning.

the middle frontal, cingulate, motor, and superior/middle temporal cortices, and the CA1/ subiculum of the hippocampus were assessed on an ordinal scale: 0 = none/rare, 1 = mild, 2 = moderate, 3 = severe/numerous. All neuropathological ratings were performed by an expert neuropathologist (J.Q.T., E.B.L.) blinded to patient genotype. We conducted ordinal logistic regression using the *MASS* package in *R* to investigate whether extent of neuronal loss rated using hematoxylin and eosin (H&E) and burden of TDP-43 pathology rated using mAbs p409/410 or 171 (Lippa *et al*, 2009; Neumann *et al*, 2009) immunohistochemistry differed according to wPRS, with covariate adjustment for age and disease duration at death.

# Data availability

All R software code generated to perform the reported analyses has been deposited online (https://github.com/pennbindlab/Polygenic ALSCognitive). Please review the associated README file for details of data access. Briefly, associated datasets can be obtained as follows:

The Clinical Research in ALS and Related Disorders for Therapeutic Development (CReATe) Consortium Phenotype–Genotype–Biomarker (PGB) study data will be deposited at the NIH-supported Data Management and Coordinating Center (DMCC) and the Database of Genotypes and Phenotypes (dbGaP) using procedures outlined by the Rare Disease Clinical Research Network (RDCRN) of the National Institutes of Health (NIH). As detailed in the patient consent process, "Only researchers with an approved study may be able to see and use your information.... Only de-identified data, which does not include anything that might directly

identify you, will be shared with study investigators and approved investigators from the general scientific community for research purposes". If you would like to access this data, please contact the CReATe Consortium at ProjectCReATe@miami.edu for a data request form.

De-identified raw T1-weighted MRI and voxel-wise cortical thickness images will be made available to researchers through an approved request pending review by the Penn Neurodegenerative Data Sharing Committee. To request access, please complete the following online data request form: https://www.pennbindlab.com/data-sharing.

The statistical code and neuropathological data from this publication have been deposited to Github: https://github.com/pennbindlab/PolygenicALSCognitive.

Expanded View for this article is available online.

## Acknowledgements

The CReATe Consortium (U54NS092091) is part of the Rare Diseases Clinical Research Network (RDCRN), an initiative of the National Center for Advancing Translational Sciences (NCATS) Office of Rare Diseases Research (ORDR). Additional research support was provided by the National Institutes of Health (NS106754, AG017586, NS092091, AG054060). The genomics sequencing was funded by St. Jude Children's Research Hospital American Lebanese Syrian Associated Charities (ALSAC), with additional support from the ALS Association for biorepository and sequencing costs (grants 17-LGCA-331 and 16-TACL-242).

## Author contributions

Study concept/design: KP, MB, CTM Acquisition, analysis, or interpretation of data: KP, MB, JW, ER, LH, VVD, DJI, LE, LM, CQ, VG, JS, TB, JR, AS, JK, EPP, CJ, JC, YS, SM, DW, EBL, JQT, PC, JG, JS, ACN, RR, GW, JPT, and CTM. Drafting/revising manuscript: KP, MB, CTM, MG, EBL, DW, VG, AN, ER, GW, WC, JS, TB.

## Conflict of interest

The following authors declare the following conflicts of interest: C.T.M. receives financial support from Biogen and has provided consulting for Axon Advisors. M.B. reports grants from National Institutes of Health, the ALS Association, the Muscular Dystrophy Association, the Centers for Disease Control and Prevention, the Department of Defense, and Target ALS during the conduct of the study; personal fees from Mitsubishi Tanabe Pharma, AveXis, Prilenia, Genentech, and Roche outside the submitted work. In addition, M.B. has a provisional patent entitled "Determining Onset of Amyotrophic Lateral Sclerosis" and serves as a site investigator on clinical trials funded by Biogen and Orphazyme. All other authors declare no competing interests.

## For more information

CReATe Consortium: https://www.rarediseasesnetwork.org/cms/create
Penn Bioinformatics in Neurodegenerative Disease Lab (Penn BiND Lab): https://www.pennbindlab.com/

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
