## [Review Process File · EMBO Molecular Medicine]

Machine learning suggests polygenic risk for cognitive dysfunction in amyotrophic lateral sclerosis

Katerina Placek, Michael Benatar, Joanne Wu, Evadnie Rampersaud, Laura Hennessy, Vivianna Van Deerlin, Murray Grossman, David Irwin, Lauren Elman, Leo McCluskey, Colin Quinn, Volkan Granit, Jeffrey Statland, Ted Burns, John Ravits, Andrea Swenson, John Katz, Erik Piro, Carlyane Jackson, James Caress, Yuen So, Samuel Maiser, David Walk, Edward Lee, John Trojanowski, Philip Cook, James Gee, Jin Sha, Adam Naj, Rosa Rademakers, CReATe Consortium, Wenan Chen, Gang Wu, J Paul Taylor, and Corey McMillan

DOI: [10.15252/emmm.202012595](https://doi.org/10.15252/emmm.202012595)

Corresponding author: Corey McMillan (mcmillac@pennmedicine.upenn.edu)

Review Timeline:

Submission Date:	24th Apr 20
Editorial Decision:	17th Jun 20
Revision Received:	18th Sep 20
Editorial Decision:	16th Oct 20
Revision Received:	27th Oct 20
Accepted:	30th Oct 20

Editor: Zeljko Durdevic

Transaction Report:

17th Jun 2020

Dear Dr. McMillan,

Thank you for the submission of your manuscript to EMBO Molecular Medicine. We have now heard back from the three referees who agreed to evaluate your manuscript. As you will see from the reports below, while referee #1 is overall supporting publication of your work, referees #2 and #3 highlight the interest of the study but also raises a number of concerns that should be addressed in a major revision of the current manuscript. Particular attention should be given to the better validation of the polygenic risk score (PRS) as a predictor of the cognitive decline in ALS/FTD patients and to the comparison of sparse canonical correlation analysis (sCCA) to the traditional regression methods for PRS generation. Addressing the reviewers' concerns in full will further be necessary for consideration of your manuscript in our journal.

Acceptance of the manuscript will entail a second round of review. Please note that EMBO Molecular Medicine encourages a single round of revision only and therefore, acceptance or rejection of the manuscript will depend on the completeness of your responses included in the next, final version of the manuscript. For this reason, and to save you from any frustrations in the end, I would strongly advise against returning an incomplete revision.

We would welcome the submission of a revised version within three months for further consideration. However, we realize that the current situation is exceptional on the account of the COVID-19/SARS-CoV-2 pandemic. Please let us know if you require longer to complete the revision.

I look forward to receiving your revised manuscript.

***** Reviewer's comments *****

Referee #1 (Remarks for Author):

Thank you for letting me see this excellent paper, reporting a study of the impact of a polygenic risk score on the presence and progression of cognitive impairment in ALS. The methodology (sparse canonical correlation analysis) appears to be a powerful approach to the analysis of multimodal datasets, and the availability of large, well-characterized samples is a major asset of the study. The results are of great relevance, as they indicate a specific effect of a set of SNPs on cognition rather than on motor impairment. The fact that they are shared with FTD strongly supports the idea of the ALS-FTD continuum. The correlation with imaging data is fully in line with these results. In the

case of neuropathology, however, the only significant effect is on the motor cortex, a point which deserves a more detailed consideration in the discussion. I suggest also to mention the limitations of the ECAS, in particular in the assessment of social cognition, an aspect related to behavioral impairment.

Referee #2 (Remarks for Author):

This study describes work to generate and validate a polygenic risk score (PRS) for predicting cognitive impairment in ALS (presence + rate of decline). The PRS is generated using a discovery cohort of 327 patients diagnosed with ALS or a related disorder. These samples have been whole genome sequenced and evaluated on multiple clinical axes, the most relevant to this story being ECAS measures of cognitive impairment. To validate the PRS, the authors utilize MRI data from a second cohort comprising 114 ALS patients and 114 controls, as well as neuropathology data from a third cohort of 88 ALS patients. The scope of the datasets is impressive and their use in a multi-modal exploration of prognostic biomarkers for ALS is without doubt of wide and considerable interest. However, some clarifications are required on the particulars of the analysis and resulting narrative.

Use of SCCA as a method to generate PRS

I like the use of SCCA in this setting. The authors dedicate much discussion to the benefits and novelty of their application, but I think some key points are missing.

1. SCCA has a long history in relating neuroimaging data to genomics. The authors could elaborate on how it has been deployed in these contexts and what is different in their use case.
2. If we consider that SCCA could be a new method for PRS generation then it is worth expending a little effort to show that it performs better than the traditional method (or at least that it is not worse). I would think a fair comparison would be to apply the traditional regression methods to estimate single variant effects in the CReATe data and construct a traditional PRS from these
3. "only one variable from the clinical dataset being chosen in each of the 10,000 iterations." So, in practice the analysis proved equivalent to assessing each clinical variable independently? Does this nullify anticipated benefits vs over traditional regularized regressions (LASSO etc)
4. The authors rightly point out that a critique of traditional PRS is "undue influence by population variance, limit their use in clinical and prognostic settings". They claim that their SCCA method avoids this, but I do not see how. The authors include PC1-2 during training but the GWAS effect sizes used in traditional PRS will also have been conditioned for PCs (and usually more than 2). The bigger problem with PRS is typically the lack of non-European samples in training datasets (Duncan, NatComm,2019). Correcting for PCs will not solve insufficient data to represent diverse human populations.

Demonstration of a "polygenic contribution to cognitive dysfunction in amyotrophic lateral sclerosis"

The authors generate a polygenic risk score (PRS) that utilizes 27 SNPs. The 27 SNPs were selected from a starting set of 45 SNPs, all of which have previously been associated with ALS and/or ALS-FTD in prior GWAS. Moreover, most of the 45 SNPs were already reported as shared risk factors for ALS / FTD and their link to cognitive symptoms is therefore established. The key narrative of the writeup is that "Our results indicate a polygenic contribution to the presence and rate of decline of cognitive". I interpret this to mean that the authors claim a novel demonstration that their 27 SNPs interact additively for "quantitative trait modification of patient phenotype". If I am mistaken could the authors clarify the claim and it's novelty? Otherwise I have the following concerns

1. I saw no clear demonstration that these variants contribute additively to modify cognitive impairment or rate of decline. The PRS has predictive value in the training dataset, but this does not immediately imply additive effects or quantitative trait modification of patient phenotype. It could be that the 27 selected SNPs comprise a collection of variants that simply tag separate endophenotypes (as demonstrated for at least C9orf72 and KIF5A), and that there is in fact no additive interplay. In-keeping with this, Fig 2 gives the impression that only a small subset of the 27 SNPs play a strong role in determining the PRS. Out of the 5 SNPs that achieve large weightings, 2 occur in the MOBP locus and these 2 SNPs have opposing directions of effect (so MOBP is a major player in determining both the largest and smallest PRS values?). Perhaps the authors could justify their claim by demonstrating that the variants have improved prognostic value when collapsed into a PRS vs when considered independently as markers for separate endophenotypes. (As it stands the reader cannot tell if the prognostic values of these known ALS-FTD SNPs is in fact reduced by collapsing into a PRS)

2. Can the authors indicate measures of LD among SNPs used in their PRS?

3. Were samples from CReATe included in the GWAS used to select variants? (if so good to acknowledge)

4. Are the primary conclusions of the study robust if the patients with "ALS related" disorders are excluded?

Validation of the PRS

1. Neuroimaging and neuropathology from independent cohorts are used to test whether the PRS captures disease relevant pathology. This has significant appeal in highlighting possible intermediate phenotypes while providing some corroboration of capturing real biological signal. However, it does not constitute a validation for how well the PRS predicts presence / rate of cognitive decline on the ECAS and such validations are standard in generating PRS. The authors include some acknowledgement of this but for the uninitiated reader it would be good to elaborate a little further on the "over-fitting" that can occur with such a study design and that true measures of model performance cannot be obtained from the same dataset used to train SCCA

2. Was the available genetic data used to exclude duplicates across the CReATe and validation cohorts?

3. Could it be clarified whether findings from the PRS analysis of the neuroimaging data are statistically significant after multiple test correction?

4. Figure 4B does not look to indicate a statistically robust linear increase in PRS with ordinal category (0-3), but the text indicates that patients with higher wPRS were x2.05 times more likely to have greater neuronal loss in motor cortex than patients with lower PGS. Does this effect size come from the ordinal logistic regression? Or does it compare all patients with PRS scores greater than the median to all patients with PRS scores below the median? Or of extreme value PRS groups? My impression is that the reader should interpret that this observation is statistically significant after multiple test correction but perhaps the authors can confirm?

5. In the CReATe dataset the authors indicate that "Polygenic score captures baseline cognition as well as longitudinal rate of cognitive decline, but not motor decline". However, in their analyses of neuronal loss/ TDP-43 pathology they indicate that they only observe effects in the motor cortex and not frontal cortex or other regions. Is there any contradiction here? Do the validation results really converge in support what was proposed in the title/ CReATe analyse?

Other comments

1. Can the authors provide summary statistics for the independent prognostic values of each of the 45 SNPs on each of the clinical variables tested? Beyond reasons already discussed, I think this would be valuable information that could be incorporated into other research

2. Could the authors include PCA plots for their samples? It would be useful to convey the homogeneity/ complexity of sample ancestry within the cohorts

Referee #3 (Remarks for Author):

The manuscript „Machine learning suggests polygenic contribution to cognitive dysfunction in amyotrophic lateral sclerosis (ALS)" by Placek and coworkers is a nicely written manuscript about a very important and timely topic about what contributes to cognitive decline in the motor devastating disease ALS. Nevertheless, I have some concerns:

major

1. Even though authors investigate reasons for „cognitive" dysfunction, they do not stratify, correlate or even properly discuss state-of-the-art subclassification of ALS according to the revised Strong criteria. While I acknowledge that the authors note that ALS-bi might need further investigation in the discussion section, I cannot accept the begin of the results saying ALS patient cohort was heterogenous also concerning cognition/cognitive test results. This is the matter why we group nowadays according to the revised Strong criteria and the study should have be drawn according to them.

2. The „biological" sample size is pretty small, the machine learning approach (to the best of my understanding) more or less an artificial multiplication of the results of these ~300 patients. Thus, it is more or less a model and not large dataset analysis. Of note, the autopsy cohort is remarkable large!

3. Even though the authors use „independent cohorts" for the neuroimaging and autopsy part, the first and necessary step would have been to try to replicate the genotype-phenotype correlations arising from the polygenic risk score also in these cohorts

4. Having said this, I'm not sure how much the knowledge about such a „polygenetic" risc score helps our further understanding of cognitive decline in ALS. Of course, it adds to the fact that cognition is not defined by a single SNPs, but this is not novel.

minor

1. Were the C9 patients in the postmortem analysis taken out of analysis?

2. Cognitive onset in the neuroimaging/autopsy cohort is defined how and how to compare this to the initial CReATe cohort, were only ALS-FTD was noted

3. Introduction: „As many as half of patients with amyotrophic lateral sclerosis (ALS) manifest progressive decline in cognition consistent with extra-motor frontal and temporal lobe neurodegeneration..."

Current understanding is that cognitive dysfunction is already present at motor disease onset and it is of current debate whether cognition also generally declines. The statement is of general and not fully proven yet!

Referee #1 (Remarks for Author):

Thank you for letting me see this excellent paper, reporting a study of the impact of a polygenic risk score on the presence and progression of cognitive impairment in ALS. The methodology (sparse canonical correlation analysis) appears to be a powerful approach to the analysis of multimodal datasets, and the availability of large, well-characterized samples is a major asset of the study. The results are of great relevance, as they indicate a specific effect of a set of SNPs on cognition rather than on motor impairment. The fact that they are shared with FTD strongly supports the idea of the ALS-FTD continuum. The correlation with imaging data is fully in line with these results. In the case of neuropathology, however, the only significant effect is on the motor cortex, a point which deserves a more detailed consideration in the discussion. I suggest also to mention the limitations of the ECAS, in particular in the assessment of social cognition, an aspect related to behavioral impairment.

We thank this reviewer for their enthusiastic comments and their constructive consideration of the anatomy of our neuroimaging and neuropathology results and importance of behavioral assessment in patients with ALS and related disorders.

Related to your point on anatomic specificity in the neuropathological data, our finding that the weighted polygenic risk score (wPRS) related to cortical thinning in regions including the prefrontal cortex and hippocampus from *in vivo* neuroimaging and the motor cortex from *post mortem* study may potentially reflect differences in two sources of sampling across the neuroimaging analysis and neuropathologic analysis.

One source of sampling differences may reflect the clinical characteristics of both cohorts. We used a retrospective analysis of neuroimaging and neuropathology data from the UPenn Biobank, which represents one of the largest repositories of multimodal data on neurodegenerative disease of its class. Our neuroimaging cohort consisted of 114 patients with ALS, 25 (17%) of whom were diagnosed with ALS-FTD and 14 (12%) of whom were diagnosed with ALS*Sci*, while our autopsy cohort consisted of 88 individuals, only 6 (7%) of whom were diagnosed with ALS-FTD and 2 (2%) of whom were diagnosed with ALS*Sci*. Thus, one possibility for our observation of the motor cortex cortical thinning relating to the wPRS at autopsy could be that a lower proportion of cases in the autopsy cohort relative to the neuroimaging cohort had impaired cognition during life. Thus, the autopsy cohort may likewise have less extensive cortical thinning in the frontal and temporal lobes due to sampling differences across cohorts. Consistent with this account, as reported in Supplementary Figures 10 and 11, a low proportion of cases in the autopsy cohort had moderate or severe neuropathological ratings in the middle frontal cortex, cingulate cortex, CA1 / subiculum, and superior/middle temporal cortex, particularly if neuronal loss in these regions are compared to the more severe levels observed in the motor cortex.

Another source of sampling differences may be technical in nature. For our neuroimaging analysis we evaluated voxelwise cortical thickness at 2mm³ resolution across the entire cortex. However, according to our established autopsy protocol for the UPenn Biobank (Toledo et al., *Alzheimer's & Dementia*, 2014) autopsy tissue is sampled from a small (e.g., 1cm) block and slides used to assess neuronal loss typically represent a 6µm section from each region of interest (e.g. middle frontal cortex, superior/middle temporal lobe). Thus, the sampled cortical tissue from neuropathological assessment only represents a microscale evaluation of total cortex.

To acknowledge these limitations we now report the proportion of patient diagnoses (e.g. ALS, ALS*Sci*, and ALS-FTD) for each UPenn cohort in Table 2. Please note these total numbers also changed as result of excluding non-independent samples between the UPenn Biobank and CReATe PGB cohorts (as suggested by Reviewer #2). We additionally revised the Discussion (page 20, paragraph 2) to state:

“Higher polygenic risk related to *in vivo* cortical thinning in the orbital prefrontal cortex, anterior cingulate cortex, premotor cortex, lateral temporal cortex, and hippocampus in a

neuroimaging cohort, and to *post-mortem* neuronal loss in sampled tissue from the motor cortex in an autopsy cohort. We speculate that the relationship to motor cortex only in the neuropathology cohort may reflect two sources of sampling differences. First, clinical characteristics differed across cohorts: 9% of the autopsy cohort had premorbid diagnoses of ALS-FTD or ALSci and 29% of the neuroimaging cohort were diagnosed with ALS-FTD or ALSci. Thus, the autopsy cohort likely had less frontal and temporal cortex neuronal loss relative to motor cortex neuronal loss. Second, the differences across analyses may reflect different scales of resolution in which neuroimaging data is analyzed at 2mm³ resolution across the entire cortex while neuropathological data is sampled at approximately 6µm. We are aware of these issues and more recently have begun to increase tissue sampling including bilateral hemisphere (Irwin et al., *Brain*, 2018; Giannini et al., *Annals of Neurology*, 2019), more extensive brain regions (Irwin et al., *Annals of Neurology*, 2016), performing digital immunohistochemistry analyses (Irwin et al., *Journal of Histochemistry and Cytochemistry*, 2015; Giannini et al., *Annals of Neurology*, 2019) and whole hemisphere post-mortem neuroimaging using 7T MRI. Thus, future studies will be able to address these sampling differences as our autopsy cohort continues to grow and our technical methods continue to improve.”

Related to your point on the limitations of not considering social cognition, we fully agree this is an important area for future research to address. Since social, behavioral and neuropsychiatric features of disease are obtained through a caregiver interviewer on the ECAS rather than direct measurements we focused our investigation on cognitive impairment. Future development of behavioral assessments that are completed by patients are necessary. We now include the following in the discussion (page 25, paragraph 2):

“We assessed patient performance on specific domains of cognition using the ECAS, this which includes a measure of social cognition counted towards the domain of executive function. Behavioral impairment on the ECAS is assessed through caregiver report (Abrahams et al., *Amyotrophic Lateral Sclerosis and Frontotemporal Dementia*, 2014), and the vast majority of neuropsychological assessments of behavior in neurodegenerative disease are based on physician or caregiver report (Simon et al., *Amyotrophic Lateral Sclerosis and Frontotemporal Dementia*, 2019). With this in mind, we chose to focus our investigation on the analysis of patient-completed assessments of cognition and motor function. Future research incorporating assessments of behavior is necessary to investigate polygenic risk for behavioral dysfunction in ALS and related disorders and to determine whether loci included in our calculated polygenic risk score additionally confer risk for behavioral dysfunction.”

Referee #2 (Remarks for Author):

This study describes work to generate and validate a polygenic risk score (PRS) for predicting cognitive impairment in ALS (presence + rate of decline). The PRS is generated using a discovery cohort of 327 patients diagnosed with ALS or a related disorder. These samples have been whole genome sequenced and evaluated on multiple clinical axes, the most relevant to this story being ECAS measures of cognitive impairment. To validate the PRS, the authors utilize MRI data from a second cohort comprising 114 ALS patients and 114 controls, as well as neuropathology data from a third cohort of 88 ALS patients. The scope of the datasets is impressive and their use in a multi-modal exploration of prognostic biomarkers for ALS is without doubt of wide and considerable interest. However, some clarifications are required on the particulars of the analysis and resulting narrative.

We would like to thank this reviewer for their thoughtful and constructive suggestions that we address below in a point-by-point response.

A. Use of SCCA as a method to generate PRS. I like the use of SCCA in this setting. The authors dedicate much discussion to the benefits and novelty of their application, but I think some key points are missing.

1. SCCA has a long history in relating neuroimaging data to genomics. The authors could elaborate on how it has been deployed in these contexts and what is different in their use case.

We agree that sCCA has a long history in imaging-genomics which we now better highlight by providing references to previous research as well as acknowledge important differences between prior applications and the current report, including the novelty of applying sCCA to polygenic risk score generation. Specifically, we re-organized and revised an Introduction paragraph to now state (page 5, paragraph 3):

“Here we employed an unsupervised machine-learning approach, sparse canonical correlation analysis (sCCA)(Witten et al., *Biostatistics*, 2009), to identify and evaluate a potential polygenic contribution to cognitive dysfunction in ALS. sCCA has previously been implemented in many contexts such as genetics (Witten et al., *Statistical Applications in Genetics and Molecular Biology*, 2009), neuroimaging-behavior studies (Avants et al., *Neuroimage*, 2014; Avants et al., *Neuroimage*, 2010), and neuroimaging-genetic studies (Du et al., *Bioinformatics*, 2019; Hao et al., *Bioinformatics*, 2017; Hu et al., *IEEE Transactions on Biomedical Engineering*, 2018), including the association of cortical thickness and white matter diffusion to FTD risk SNPs (McMillan et al., *Neurobiology of Aging*, 2014). For the first time, we leverage sCCA as a data-driven tool to facilitate generation of a polygenic risk score. Specifically, sCCA can be leveraged to identify variants by employing sparsity to select maximally contributing variants and to assign corresponding weights based on model contribution with minimal *a priori* assumptions. This contrasts with traditional approaches to constructing polygenic scores that rely on the use of existing GWAS statistics to select variants and assign weights, which can be challenging if the original GWAS statistics are based on case-control associations rather than current neuropsychological outcome of interest.”

We also have added the following statement to the Discussion (page 21, paragraph 2):

“While sCCA has been widely applied to genotype-phenotype studies (Witten et al., *Statistical Applications in Genetics and Molecular Biology*, 2009; Parkhomenko et al., *Statistical Applications in Genetics and Molecular Biology*, 2009), including neuroimaging-genetic studies (Avants et al., *Neuroimage*, 2014; Avants et al., *Neuroimage*, 2010; Du et al., *Bioinformatics*, 2019; Hao et al., *Bioinformatics*, 2017; Hu et al., *IEEE Transactions on Biomedical Engineering*, 2018; McMillan et al., *Neurobiology of*

Aging, 2014), we are unaware of prior applications using sCCA to define a polygenic score based on rich clinical phenotypic and biomarker data.”

2. If we consider that SCCA could be a new method for PRS generation then it is worth expending a little effort to show that it performs better than the traditional method (or at least that it is not worse). I would think a fair comparison would be to apply the traditional regression methods to estimate single variant effects in the CReATe data and construct a traditional PRS from these

This is an excellent suggestion and we considered reporting something similar to this in the original submission but opted not to due to challenges with such an implementation. In particular, traditional approaches to the generation of polygenic risk scores include selection of SNPs from a single GWAS and weighting the selected SNPs based on the GWAS odds ratio statistics for the phenotype associations, typically case vs. control. In the current study, we diverged from this traditional approach in two ways. First, we investigated 45 SNPs selected from two different hypothesis-driven sources: a genome-wide conjunction analysis for FTD and ALS (Karch et al., *JAMA Neurology*, 2018) and the largest case-control ALS GWAS to date (Nicolas et al., *Neuron*, 2018). Second, by selecting these SNPs, their associated weights from the prior conjunction analysis and case-control GWAS would not be applicable since they were generated using a “phenotype” distinct from our research question related to cognitive outcomes. With this in mind, we are unable to calculate a traditional polygenic risk score.

Nonetheless, in an effort to compare our sCCA approach to alternative method we calculated an unweighted polygenic risk score from the genetic variables selected from our sCCA analysis by computing an unweighted sum of allele dosages for each genetic variable. This approach is described in the revised Methods (page 13, paragraph 1):

“We also calculated an unweighted polygenic risk score (uPRS) by computing an unweighted sum of allele dosage for each individual genetic variable selected from sCCA modeling.”

We now also report associations with the uPRS relative to 1) ECAS performance in the CReATe cohort, 2) *in vivo* cortical thickness in the UPenn Biobank neuroimaging cohort, and 3) *post mortem* severity of cortical thinning and burden of TDP-43 neuropathology UPenn Biobank autopsy cohort. We did not find any significant relationship between the uPRS and any of these measures, as reported in the Results (pages 13-15):

Page 13, paragraph 1: “We observed no statistically significant relationship between the uPRS and adjusted baseline estimates of performance on ALS-Specific, Executive Function, Language, and ECAS Total scores (all p values $> .2$).”

Page 14, paragraph 1: “We observed no statistically significant relationship between the uPRS and adjusted rate of decline on any clinical measure (all p values $> .9$).”

Page 16, paragraph 1: “We observed no statistically significant relationship between the uPRS and cortical thickness in any region (not shown).”

Page 17, paragraph 1: “We also observed no statistically significant associations between the uPRS and neuronal loss or TDP-43 pathology in region (all p values > 0.19 ; not shown).”

We interpret these findings to suggest that the weights derived from sCCA meaningfully define the relationship between genetic variation and quantitative phenotypic differences in the CReATe PGB and UPenn cohorts with regards to cognitive performance and disease neuroanatomy. We include an interpretation of these findings in the Discussion (page 21, paragraph 2):

“The positive or negative direction of model-derived weights is potentially biologically informative, and could reflect ‘risk’ (i.e. positive weight) or ‘protective’ (i.e. a negative weight) effects. We evaluated the wPRS using model-derived weights relative to a uPRS derived created by computing an unweighted sum of allele dosages for each genetic variable. Our observation that the uPRS did not relate to cognitive or clinical performance in the CReATe PGB cohort or to neuroimaging or neuropathology in the UPenn Biomarker cohorts suggests that that the weights derived from sCCA meaningfully define the relationship between genetic variation and quantitative phenotypic differences in the CReATe PGB and UPenn cohorts with regards to cognitive performance and disease neuroanatomy. ”

3. *"only one variable from the clinical dataset being chosen in each of the 10,000 iterations." So, in practice the analysis proved equivalent to assessing each clinical variable independently? Does this nullify anticipated benefits vs over traditional regularized regressions (LASSO etc)*

We acknowledge this reviewer’s concern regarding the selection of clinical variables by sCCA modeling. We used a gridsearch of 100 possible combinations of L1 parameters ranging in increments from 0.1 (most stringent selection of 10% of features) to 1.0 (least stringent selection of all 100% of features) for the clinical and genetic datasets, and found that the optimal L1 parameter for the clinical dataset was 0.1 (the most stringent L1 parameter) and the optimal L1 parameter for the genetic dataset was 0.6. While the usage of these L1 parameters resulted in one clinical variable being chosen in each of the 10,000 iterations, this approach differs from LASSO in several key ways. First, in LASSO, the dependent variable is pre-specified by the researcher; however, in the current study, sCCA modeling was used to select a single clinical variable in each model iteration. This use of sCCA to select a single clinical variable, rather than pre-specifying a clinical variable resulted in the minimization of experimenter bias. Second, 11 LASSO models would need to be run in order to investigate each of the 11 clinical variables as a dependent variable; SCCA offers advantages over this approach with regards to multiple testing considerations because it allows the concurrent consideration of all 11 clinical variables in each model iteration. We address this concern by clarifying the quoted statement in the Results (page 10, paragraph 1):

“This differs from other regularization techniques (e.g. LASSO), as the variable from the clinical dataset was selected by sCCA modeling in each iteration rather than being experimenter-selected prior to analysis. Importantly, the use of sCCA also minimizes the necessity for multiple comparisons corrections, since all variables can be tested in a single model, and therefore reduces the potential of a Type II false-negative error common in genomics studies related to rejection of a true effect due to overly stringent correction of multiple comparisons.”

4. *The authors rightly point out that a critique of traditional PRS is "undue influence by population variance, limit their use in clinical and prognostic settings". They claim that their SCCA method avoids this, but I do not see how. The authors include PC1-2 during training but the GWAS effect sizes used in traditional PRS will also have been conditioned for PCs (and usually more than 2). The bigger problem with PRS is typically the lack of non-European samples in training datasets (Duncan, NatComm,2019). Correcting for PCs will not solve insufficient data to represent diverse human populations.*

We agree with this reviewer in recognizing the importance of acknowledging limitations in racial and ethnic diversity in the creation of polygenic scores and in natural history studies more broadly, and the critical need for increased inclusivity and representation for individuals from diverse racial and ethnic groups.

We altered language in the Methods and Results to more accurately state that our inclusion of PC1 and PC2 is an attempt to account for differences in population substructure:

Results, page 9, paragraph 2: "We included the first two principal components from a PCA conducted in the PGB cohort and binary variables for sex, *C9ORF72* repeat expansion status, and other mutation status (e.g. *SOD1*) in this dataset in an effort to account for inter-individual genetic differences in population structure, sex, and mutation status."

Methods, page 30, paragraph 1: "In an attempt to account for population substructure, we additionally derived the first two principal components scores for each in the CReATe PGB cohort using principal components analysis (PCA) implemented using Eigenstrat".

We now include the following statement in the Discussion, citing the study mentioned by the reviewer, to help address this limitation (page 24, paragraph 1):

"Previous critiques of polygenic scores argue that three factors limit their use in clinical and prognostic settings: (1) calculation based on GWAS-defined odds ratios for univariate risk loci; (2) undue influence by population variance; and (3) predominant use of samples of European ancestry (Wald & Old, *Genetics in Medicine*, 2019; Duncan et al., *Nature Communications*, 2019). In an attempt to mitigate these potential confounds, we based our computation of the wPRS on model-selected parameters derived from an analysis including all genetic variants and, in addition, covariates for genetic mutation status and sex in an effort to account for multivariate genetic relationships. We also included the first two principal components in our model from a PCA conducted in the CReATe PGB cohort in an effort to account for differences in population substructure (Price et al., *Nature Genetics*, 2006). While we used the first two principal components in an effort to account for population substructure, this is a complex issue to resolve and future studies with more diverse cohorts to investigate potential substructure bias are necessary. The current investigation utilized existing data from natural history studies that were predominantly comprised of individuals of European ancestry; however, increased representation of diverse racial and ethnic groups in future investigations of polygenic risk for cognitive impairment in ALS are necessary in order to ensure generalizability to diverse populations."

B. Demonstration of a "polygenic contribution to cognitive dysfunction in amyotrophic lateral sclerosis". The authors generate a polygenic risk score (PRS) that utilizes 27 SNPs. The 27 SNPs were selected from a starting set of 45 SNPs, all of which have previously been associated with ALS and/or ALS-FTD in prior GWAS. Moreover, most of the 45 SNPs were already reported as shared risk factors for ALS / FTD and their link to cognitive symptoms is therefore established. The key narrative of the writeup is that "Our results indicate a polygenic contribution to the presence and rate of decline of cognitive". I interpret this to mean that the authors claim a novel demonstration that their 27 SNPs interact additively for "quantitative trait modification of patient phenotype". If I am mistaken could the authors clarify the claim and it's novelty? Otherwise I have the following concerns

We address each of these concerns below in detail.

*1. I saw no clear demonstration that these variants contribute additively to modify cognitive impairment or rate of decline. The PRS has predictive value in the training dataset, but this does not immediately imply additive effects or quantitative trait modification of patient phenotype. It could be that the 27 selected SNPs comprise a collection of variants that simply tag separate endophenotypes (as demonstrated for at least *C9orf72* and *KIF5A*), and that there is in fact no additive interplay. In-keeping with this, Fig 2 gives the impression that only a small subset of the 27 SNPs play a strong role in determining the PRS. Out of the 5 SNPs that achieve large weightings, 2 occur in the *MOBP* locus and these 2 SNPs have opposing directions of effect (so*

MOBP is a major player in determining both the largest and smallest PRS values?). Perhaps the authors could justify their claim by demonstrating that the variants have improved prognostic value when collapsed into a PRS vs when considered independently as markers for separate endophenotypes. (As it stands the reader cannot tell if the prognostic values of these known ALS-FTD SNPs is in fact reduced by collapsing into a PRS)

We appreciate your attention to detail about the relative independence vs. additive nature of selected SNPs, the predominant strength of 5 SNPs in our resulting PRS that included 27 selected SNPs, and the directionality of associations. We have addressed each of these points in the manuscript.

Related to an “additive effect” we agree that just because 27 SNPs were selected they may not have an additive or cumulative impact, but rather collectively contribute to endophenotypes, and we specifically avoid the term “additive” when describing the combination of SNPs for the PRS. We have revised the manuscript throughout to refrain from implying that these associations are necessarily “additive” in nature. We additionally revised the Discussion to state the following (page 22, paragraph 1).

“While our sCCA modeling selected 27 SNPs in addition to sex and *C9ORF72* mutation status and we used model-derived weights to calculate a wPRS, we are unable to determine in the current study what the collective contribution of these SNPs are to modifying cognitive phenotypes. For example, these could be additive in nature, such that increased risk allele dosage increases risk for impaired cognition, or the selected SNPs could act independently in disease modification. As often is the case, future functional studies are required to identify the mechanistic relationship between SNP associations and cognitive phenotype.”

To address the independent contribution of SNPs and their strength and directionality, we incorporated univariate analyses of each clinical variable and SNP genotype into the manuscript and report this in the Methods (page 31, paragraph 3):

“In addition to the linear mixed-effects models described above, we also conducted a second series of linear-mixed effects models to investigate fixed effects of each of the 45 SNPs on each of the 11 clinical measures (i.e. all ECAS scores, ALSFRS-R, and UMN and LMN burden scores), independently; this resulted in a total of 495 models. We again used the *nlme* package in R and each model was fit using maximum likelihood. In addition to each SNP, we included age at baseline visit (in years), lag between age of symptom onset and age at baseline visit (in years), college education (yes / no), bulbar onset (yes / no) and visit time-point (in months) as fixed effects, and we included individual-by-visit time-point as a random effect.”

We now report in the Results and include a supplementary figure (*Supplementary Figure 8*) demonstrating that the five SNPs achieving the strongest median weights from bootstrapped sCCA modeling do emerge in this univariate analysis but do not survive multiple comparisons corrections (page 14, paragraph 2).

“In *post hoc* analyses, we investigated whether SNPs also contribute individually to rate of decline on clinical measures. We conducted LME modeling of the original longitudinal data to investigate fixed effects of each of the 45 SNPs on each of the 11 clinical measures (i.e. all ECAS scores, ALSFRS-R, and UMN and LMN burden scores), independently. We did not observe any effects that survived corrections for multiple comparisons. However, we observed that the SNPs achieving the five largest median weights from bootstrapped sCCA modeling (rs1768208, rs538622, rs10143310, rs7224296, rs9820623) also independently related to performance on the ECAS ALS-Specific and Total scores (all uncorrected $p < .05$).”

Given that the univariate analyses did not survive multiple comparisons we also added the following to the Discussion (page 22, paragraph 1):

“*Post hoc* investigation of independent SNP effects on longitudinal cognitive performance revealed that the SNPs achieving the five largest median weights from bootstrapped sCCA modeling also relate to longitudinal cognitive performance; however these effects did not survive correction for multiple comparisons. By its nature, this *post hoc* investigation considered each SNP as independent from other SNPs and each clinical measure as independent from other clinical measures and thus did not account for more complex collective contribution of SNPs to cognitive phenotypes. As often is the case, future functional studies are required to identify the mechanistic relationship between SNP associations and cognitive phenotype. Nonetheless, our results support the consideration of sCCA as a promising method to identify collective combinations of SNPs and cognitive phenotypes and to direct research efforts towards model-selected variants.”

2. Can the authors indicate measures of LD among SNPs used in their PRS?

We thank this reviewer for their interest in understanding how LD contributes to our calculated PRS. We report D' and R^2 derived from LDLink using European populations for pairs of SNPs on the same chromosomes in the Supplementary Table 2. Critically, while our analyses included 5 SNPs in high LD ($D' > 0.8$), sCCA notably is able to accommodate highly-correlated features, including genetic variants. We revised the Methods to state (Page 30, paragraph 2):

“An assessment of LD revealed that 5 of our 45 hypothesized SNPs were in high LD with one another ($D' > 0.8$; *Supplementary Table 2*), but we included these high LD SNPs in our investigation since sCCA is able to accommodate highly correlated features (Witten et al., *Biostatistics*, 2009).”

We additionally now report a comparison of a wPRS that includes and does not include the high LD SNPs (page 14, paragraph 3):

“We also conducted *post hoc* analyses to investigate whether the inclusion of SNPs in high linkage disequilibrium (LD) influence the magnitude and direction of the wPRS we re-ran bootstrapped sCCA analyses using 10,000 iterations excluding the 5 SNPs in high LD (i.e. based on the cutoff of $R^2 > 0.5$) and recalculated the wPRS in the CReATe cohort. This revealed a strong linear relationship between both wPRS models (Pearson's $R = 0.90$ (95% CI: 0.87, 0.91), $p < 2.2 \times 10^{-16}$; *Supplementary Figure 7*) and thus LD of a subset of SNPs is unlikely to be a driver of our observed polygenic associations.”

3. Were samples from CReATe included in the GWAS used to select variants? (if so good to acknowledge)

We appreciate your attention to detail and revised the Methods to clarify (page 27, paragraph 1):

“A subset of 155 CReATe PGB cases were previously included in the replication cohort of the ALS case-control GWAS (Nicolas et al., *Neuron*, 2018).”

4. Are the primary conclusions of the study robust if the patients with “ALS related” disorders are excluded?

We appreciate this reviewer's concern for the inclusion of patients in the CReATe PGB cohort who were diagnosed with “ALS related” disorders, a term encompassing patients with primary lateral sclerosis (PLS) and progressive muscular atrophy (PMA). We included these individuals in our investigation for two primary reasons: 1) Individuals initially diagnosed with PLS or PMA may

eventually progress to diagnostic criteria for ALS (e.g. Kim et al., *Neurology*, 2009) and 2) Individuals with PLS and PMA show similar profiles of cognitive dysfunction to individuals with ALS (de Vries et al., *JNNP*, 2019). Nonetheless, to address this concern we re-ran bootstrapped sCCA modeling as originally reported on a subset of the CReATe PGB excluding 35 patients with ALS-related disorders (N=22 PLS, N=13 PMA). We describe this revised approach in the Results:

Page 7, paragraph 1: “We included a spectrum of ALS and related disorder cases in an effort to account for the possibility that a subset of PLS or PMA cases may evolve into ALS (Kim et al., *Neurology*, 2009) and can have similar cognitive profiles of cognitive dysfunction to ALS (de Vries, *JNNP*, 2019).

Page 12, paragraph 1: “To evaluate whether our observed sCCA model was impacted by inclusion of patients with disorders related to ALS (i.e. PLS, PMA), we compared the median weights for genetic features and the percentage of times selected for clinical features from sCCA modeling using the entire CReATe PGB cohort (i.e. with PLS and PMA included) to those obtained from sCCA modeling using a subset of the CReATe PGB cohort that excluded patients with PLS and PMA. sCCA modeling that excluded patients with PLS and PMA resulted in the most frequent selection of the ECAS Total, ALS-Specific, Executive Function, and Language scores (*Supplementary Figure 6A*), similar to results obtained in the entire cohort. Furthermore, sCCA modeling that excluded patients with PLS and PMA resulted in the same selection of genetic variables as in sCCA modeling of the entire cohort, and achieved similar direction and strength of weights (*Supplementary Figure 6B*). This demonstrates that the inclusion of disorders related to ALS does not potentially confound our observations.”

We also now include a sentence in the Discussion addressing these findings (page 23, paragraph 2):

“While we focus on ALS and FTD risk variants and demonstrate that the inclusion of related disorders (i.e. PLS, PMA) does not confound our observed cognitive and genetic associations, future work should also incorporate variants associated with risk for disorders related to ALS and specifically test the application of polygenic associations within PLS and PMA.”

C. Validation of the PRS

1. Neuroimaging and neuropathology from independent cohorts are used to test whether the PRS captures disease relevant pathology. This has significant appeal in highlighting possible intermediate phenotypes while providing some corroboration of capturing real biological signal. However, it does not constitute a validation for how well the PRS predicts presence / rate of cognitive decline on the ECAS and such validations are standard in generating PRS. The authors include some acknowledgement of this but for the uninitiated reader it would be good to elaborate a little further on the “over-fitting” that can occur with such a study design and that true measures of model performance cannot be obtained from the same dataset used to train SCCA

We appreciate your notion that our multimodal analyses provide some corroboration of capturing real biological signal and also agree with potential dangers of model over-fitting, especially with regards to machine learning approaches. In an effort to minimize over-fitting we 1) used bootstrapped sCCA modeling on subsets of 75% of the CReATe cohort and 2) report median weights over 10,000 permutations of bootstrapped sCCA modeling rather than a single “top model”. We revised the following text of the Discussion to highlight the potential limitations of this approach (page 25, paragraph 2):

“Although the current study demonstrates converging, multimodal evidence for polygenic risk, replication in additional cohorts with larger sample sizes that allow for robust cross-validation is warranted. Notably, machine-learning methods have the tendency to over-fit data and produce estimates that do not generalize to different data sets. However, alternative datasets for ALS that contain detailed genotyping and cognitive phenotyping

are currently lacking and the CReATe PGB cohort represents the largest of its kind. In the absence of an alternative dataset to minimize over-fitting, we employed a bootstrapping procedure and generated a final sCCA model based on median weights across permutations rather than selecting a single “top model”. We additionally demonstrate converging, multimodal evidence for polygenic risk in independent neuroimaging and neuropathology biomarker cohorts in an effort to provide corroboration that we are detecting a true biological signal. However, future research is necessary to determine the predictive potential and generalizability of our proposed polygenic risk score in ALS patients. We furthermore hope that this demonstration motivates the collection of additional genotyping data and longitudinal cognitive evaluation using the ECAS in additional large-scale patient cohorts.”

2. Was the available genetic data used to exclude duplicates across the CReATe and validation cohorts?

We thank the reviewer for this attention to detail and agree with the need to exclude any individuals from the converging evidence cohorts at UPenn who were also included in the CReATe PGB cohort. By linking IDs across cohorts, in the absence of WGS data in the UPenn cohorts, we identified 1 individual from the UPenn Biobank autopsy cohort and 22 individuals in the UPenn neuroimaging cohort that were also in the CReATe PGB cohort. We therefore excluded these 23 individuals from the neuroimaging and neuropathological analyses and revised the Methods to state:

Page 33, paragraph 2: “We retrospectively evaluated 90 patients with ALS and 90 healthy controls matched for age, sex, and education from the UPenn Biobank.... Inclusion criteria for ALS patients consisted of the following: lack of participation in the CReATe PGB cohort....”

Page 35, paragraph 2: “We evaluated brain tissue samples from 87 ALS autopsy cases identified from the UPenn Biobank.... Inclusion criteria consisted of the following: lack of participation in the CReATe PGB cohort....”

We then re-ran all neuroimaging and neuropathology analyses after excluding these individuals, and report the updated analyses throughout the Results. Critically, this did not impact our previously reported associations.

3. Could it be clarified whether findings from the PRS analysis of the neuroimaging data are statistically significant after multiple test correction?

Clear reporting of statistical thresholds in a scientific manuscript is essential to the interpretation of results.

We now report statistical thresholds in the Results section (Page 15, paragraph 2):

“Nonparametric modeling using 10,000 random permutations revealed extensive reduction of cortical thickness bilaterally in the frontal and temporal cortices of patients relative to controls (threshold-free cluster enhancement, FWE corrected $p < 0.05$)....

Nonparametric modeling using 10,000 random permutations with adjustments for potential confounds in age, disease duration, and scanning acquisition revealed that a higher wPRS (i.e. greater risk) associated with greater reduction of cortical thickness in regions including the orbital prefrontal cortex, anterior cingulate cortex, premotor cortex, lateral temporal cortex, and hippocampus that survived uncorrected p value of 0.01 and a cluster extent threshold of 10 voxels (*Figure 4A; Supplementary Table 3*).”

In the Methods section, we also report (Page 38, paragraph 2):

“First, we used *randomise* set to 10,000 permutations to identify reduced cortical thickness in ALS patients relative to healthy controls. We constrained this analysis using an explicit mask restricted to high probability cortex (>0.4) and reported clusters that survive $p < 0.05$ threshold-free cluster enhancement (Smith et al., *NeuroImage*, 2009) corrected for family-wise error (FWE).

Next, we again used *randomise* set to 10,000 permutations to identify regions of reduced cortical thickness associated with wPRS in ALS patients, constraining analysis to an explicit mask defined by regions of reduced cortical thickness in ALS patients relative to controls (see above). The statistical model for this analysis included covariate adjustment for age, disease duration, and scanner acquisition. We report clusters that survive uncorrected $p < 0.01$ with a cluster extent threshold of 10 voxels; we employ an uncorrected threshold to minimize the chance of Type II error (not observing a true result).”

4. *Figure 4B does not look to indicate a statistically robust linear increase in PRS with ordinal category (0-3), but the text indicates that patients with higher wPRS were x2.05 times more likely to have greater neuronal loss in motor cortex than patients with lower PRS. Does this effect size comes from the ordinal logistic regression? Or does it compare all patients with PRS scores greater than the median to all patients with PRS scores below the median? Or of extreme value PRS groups? My impression is that the reader should interpret that this observation is statistically significant after multiple test correction but perhaps the authors can confirm?*

We thank this reviewer for their interest in clarifying the effect of the wPRS on neuronal loss in the autopsy cohort and apologize for lack of clarity in reporting this statistic based on continuous data. We report the effect from ordinal logistic regression as the odds associated with wPRS for ordinal score of magnitude of neuronal loss. This odds ratio comes from a multivariable model that includes wPRS, age, and disease duration at death as covariates. We also thank the reviewer regarding the clarity of statistical reporting. Our reported result is not statistically significant after multiple comparisons correction using family-wise error adjustment. We now clearly state in the Results as follows (page 16, paragraph 2):

“We conducted ordinal logistic regression with covariate adjustment for age at death and disease duration and found that as patients’ wPRS increases, their odds of greater neuronal loss in the motor cortex also increases (OR = 1.98; 95% CI: 1.01, 3.96; uncorrected $p = 0.049$; *Figure 4B*); older age at death and longer disease duration were not found to statistically significantly influence the odds for greater neuronal loss ($p > 0.05$).”

5. *In the CReATe dataset the authors indicate that "Polygenic score captures baseline cognition as well as longitudinal rate of cognitive decline, but not motor decline". However, in their analyses of neuronal loss/ TDP-43 pathology they indicate that they only observe effects in the motor cortex and not frontal cortex or other regions. Is there any contradiction here? Do the validation results really converge in support what was proposed in the title/ CReATe analyses?*

We thank this reviewer for their consideration of the anatomy of our neuroimaging and neuropathology results and for acknowledging the importance of behavioral assessment in patients with ALS and related disorders.

Our finding that the wPRS related to cortical thinning in regions including the prefrontal cortex and hippocampus from *in vivo* neuroimaging and the motor cortex from *post mortem* study may potentially reflect differences in two sources of sampling across the neuroimaging analysis and neuropathologic analysis.

One source of sampling differences may reflect the clinical characteristics of both cohorts. We used a retrospective analysis of neuroimaging and neuropathology data from the UPenn Biobank, which represents one of the largest repositories of multimodal data on neurodegenerative disease of its class. Our neuroimaging cohort consisted of 114 patients with ALS, 25 (17%) of whom were diagnosed with ALS-FTD and 14 (12%) of whom were diagnosed with ALS*Sci*, while our autopsy cohort consisted of 88 individuals, only 6 (7%) of whom were diagnosed with ALS-FTD and 2 (2%) of whom were diagnosed with ALS*Sci*. Thus, one possibility for our observation of the motor cortex cortical thinning relating to the wPRS at autopsy could be that a lower proportion of cases in the autopsy cohort relative to the neuroimaging cohort had impaired cognition during life. Thus, the autopsy cohort may likewise have less extensive cortical thinning in the frontal and temporal lobes due to sampling differences across cohorts. Consistent with this account, as reported in Supplementary Figures 10 and 11, a low proportion of cases in the autopsy cohort had moderate or severe neuropathological ratings in the middle frontal cortex, cingulate cortex, CA1 / subiculum, and superior/middle temporal cortex, particularly if neuronal loss in these regions are compared to the more severe levels observed in the motor cortex.

Another source of sampling differences may be technical nature. For our neuroimaging analysis we evaluated voxelwise cortical thickness at 2mm^3 resolution across the entire cortex. However, according to our established autopsy protocol for the UPenn Biobank (Toledo et al., *Alzheimer's & Dementia*, 2014) autopsy tissue is sampled from a small (e.g., 1cm) block and slides used to assess neuronal loss typically represent a $6\mu\text{m}$ section from each region of interest (e.g. middle frontal cortex, superior/middle temporal lobe). Thus, the sampled cortical tissue from neuropathological assessment only represents a microscale evaluation of total cortex.

To acknowledge these limitations we now report the proportion of patient diagnoses (e.g. ALS, ALS*Sci*, and ALS-FTD) for each UPenn cohort in Table 2. Please note these total numbers also changed as result of excluding non-independent samples between the UPenn and CReATe cohorts (as suggested by Reviewer #2). We additionally revised the Discussion (page 20, paragraph 2) to state:

“Higher polygenic risk related to *in vivo* cortical thinning in the orbital prefrontal cortex, anterior cingulate cortex, premotor cortex, lateral temporal cortex, and hippocampus in a neuroimaging cohort, and to *post-mortem* neuronal loss in sampled tissue from the motor cortex in an autopsy cohort. We speculate that the relationship to motor cortex only in the neuropathology cohort may reflect two sources of sampling differences. Higher polygenic risk related to *in vivo* cortical thinning in the orbital prefrontal cortex, anterior cingulate cortex, premotor cortex, lateral temporal cortex, and hippocampus in a neuroimaging cohort, and to *post-mortem* neuronal loss in sampled tissue from the motor cortex in an autopsy cohort. We speculate that the relationship to motor cortex only in the neuropathology cohort may reflect two sources of sampling differences. First, clinical characteristics differed across cohorts: 9% of the autopsy cohort had premorbid diagnoses of ALS-FTD or ALS*Sci* and 29% of the neuroimaging cohort were diagnosed with ALS-FTD or ALS*Sci*. Thus, the autopsy cohort likely had less frontal and temporal cortex neuronal loss relative to motor cortex neuronal loss. Second, the differences across analyses may reflect different scales of resolution in which neuroimaging data is analyzed at 2mm^3 resolution across the entire cortex while neuropathological data is sampled at approximately $6\mu\text{m}$. We are aware of these issues and more recently have begun to increase tissue sampling including bilateral hemisphere (Irwin et al., *Brain*, 2018; Giannini et al., *Annals of Neurology*, 2019), more extensive brain regions (Irwin et al., *Annals of Neurology*, 2016), performing digital immunohistochemistry analyses (Irwin et al., *Journal of Histochemistry and Cytochemistry*, 2015; Giannini et al., *Annals of Neurology*, 2019) and whole hemisphere post-mortem neuroimaging using 7T MRI. Thus, future studies will be able to address these sampling differences as our autopsy cohort continues to grow and our technical methods continue to improve.”

D. Other comments

1. Can the authors provide summary statistics for the independent prognostic values of each of the 45 SNPs on each of the clinical variables tested? Beyond reasons already discussed, I think this would be valuable information that could be incorporated into other research.

As responded in your prior comment B.1 we now report univariate associations of each 45 SNPs on each of the 11 clinical variables in Supplementary Figure 8.

2. Could the authors include PCA plots for their samples? It would be useful to convey the homogeneity/ complexity of sample ancestry within the cohorts

We appreciate this reviewer's request for PCA plots for the 327 patients included in the CReATe PGB cohort, and now include a PC1 x PC2 plot in Supplementary Figure 1. As can be observed the vast majority of cases cluster together consistent with a White European cohort and small proportion of 10-20 cases extend along an axis likely reflecting African/Admixed and/or Asian ancestry.

Referee #3 (Remarks for Author):

The manuscript „Machine learning suggests polygenic contribution to cognitive dysfunction in amyotrophic lateral sclerosis (ALS)” by Placek and coworkers is a nicely written manuscript about a very important and timely topic about what contributes to cognitive decline in the motor devastating disease ALS. Nevertheless, I have some concerns:

Thank you for your constructive comments and appreciation that this manuscript is nicely written as well as a very important and timely topic.

A. Major Concerns

1. Even though authors investigate reasons for „cognitive” dysfunction, they do not stratify, correlate or even properly discuss state-of-the-art subclassification of ALS according to the revised Strong criteria. While I acknowledge that the authors note that ALS-bi might need further investigation in the discussion section, I cannot accept the begin of the results saying ALS patient cohort was heterogeneous also concerning cognition/cognitive test results. This is the matter why we group nowadays according to the revised Strong criteria and the study should have been drawn according to them.

We agree that clinically-defined subclassifications of ALS according to the revised Strong criteria are important for facilitating the interpretation of our reported findings and better understanding the clinical/cognitive characteristics of our cohorts. However, there are also some limitations with Strong criteria: (1) boundaries to define cognitive impairment are arbitrary and require appropriate normative data to classify “impaired” on each domain that are currently lacking; (2) even with a Strong criteria category like ALSci there is heterogeneity, for example, it can include executive dysfunction, language dysfunction or a combination of the two; and (3) behavioral assessments are informant based and thus not always reliable. Therefore, to focus on patient-derived data we report the ALS, ALSci, and ALS-FTD Strong criteria categories, but not ALSbi, using the following approaches to provide a descriptive summary of cohort patient characteristics, but continue to analyze data as a matrix of continuous ECAS values that captures the complex heterogeneity observed across the ALS frontotemporal dementia spectrum. Specific edits include:

Page 27, paragraph 1: (CReATe PGB cohort): “Presence of ALS with cognitive impairment (ALSci) was assessed at baseline using the ECAS according to established criteria (Strong et al., *Amyotrophic Lateral Sclerosis and Frontotemporal Dementia*, 2017), operationalized as baseline performance on Executive Function, Verbal Fluency, or Language subscores at or below normative cutoff scores (Abrahams et al., *Amyotrophic Lateral Sclerosis and Frontotemporal Dementia*, 2014).”

Page 34, paragraph 3: (UPenn Biobank neuroimaging cohort): “All patients were diagnosed with ALS by a board-certified neurologist (L.E., L.M., M.G., D.I.) using revised El Escorial criteria and assessed for ALS frontotemporal spectrum disorder using established criteria; those patients enrolled in research prior to 2017 were retrospectively evaluated through chart review.”

Page 35, paragraph 2: (UPenn Biobank autopsy cohort): “During life, all patients were diagnosed with ALS by a board-certified neurologist (L.E., L.M., M.G., D.I.) using revised El Escorial criteria and assessed for ALS frontotemporal spectrum disorder using established criteria; those patients enrolled in research prior to 2017 were retrospectively evaluated through chart review.”

We also have revised Table 1, Table 2, Figure 1A and Figure 1C to include ALS, ALSci, and ALS-FTD as well as PLS and PMA.

2. The „biological” sample size is pretty small, the machine learning approach (to the best of my understanding) more or less an artificial multiplication of the results of these ~300 patients. Thus,

it is more or less a model and not large dataset analysis. Of note, the autopsy cohort is remarkable large!

We agree that sample size is an important consideration in any machine learning approach. While small relative to other big genomic series, the CReATe PGB cohort (N=339) represents the largest longitudinal phenotype-genotype dataset of its class with detailed cognitive and clinical assessment and available whole genome sequencing data. In our machine learning analyses, we used bootstrapping in an effort to 1) avoid model over-fitting to the CReATe PGB cohort data, and 2) evaluate model performance using randomly-permuted data in the absence of an independent replication cohort (see Response to point #3 below). Furthermore, we feel that our observation of biological signals in independent modalities including neuroimaging and neuropathology provide some corroboration that our observations have some generalizability. Nonetheless, we revised the manuscript to acknowledge sample size limitations in the Discussion (Page 25, paragraph 2):

“Although the current study demonstrates converging, multimodal evidence for polygenic risk, replication in additional cohorts with larger sample sizes that allow for robust cross-validation is warranted. Notably, machine-learning methods have the tendency to over-fit data and produce estimates that do not generalize to different data sets. However, alternative datasets for ALS that contain detailed genotyping and cognitive phenotyping are currently lacking and the CReATe PGB cohort represents the largest of its kind. In the absence of an alternative dataset to minimize over-fitting, we employed a bootstrapping procedure and generated a final sCCA model based on median weights across permutations rather than selecting a single “top model”. We additionally demonstrate converging, multimodal evidence for polygenic risk in independent neuroimaging and neuropathology biomarker cohorts in an effort to provide corroboration that we are detecting a true biological signal. However, future research is necessary to determine the predictive potential and generalizability of our proposed polygenic risk score in ALS patients. We furthermore hope that this demonstration motivates the collection of additional genotyping data and longitudinal cognitive evaluation using the ECAS in additional large-scale patient cohorts.”

3. Even though the authors use „independent cohorts“ for the neuroimaging and autopsy part, the first and necessary step would have been to try to replicate the genotype-phenotype correlations arising from the polygenic risk score also in these cohorts.

We agree with this reviewer regarding the importance for validation of our observed genotype-phenotype relationship in the CReATe PGB cohort in additional, independent cohorts. However, given the lack of additional ALS patient cohorts with longitudinal cognitive evaluation on the ECAS, we were unable to validate our model in independent cohorts. To address this, we investigated whether a polygenic risk score derived from the median weights of bootstrapped sCCA relate to neuroimaging and neuropathology in the UPenn cohorts. Nonetheless, we also agree that true validation of our sCCA model and derived polygenic risk score is necessary in additional patient cohorts with longitudinal ECAS performance and targeted genotyping. We have edited our discussion of the need for cross-validation in the Discussion as follows (Page 25, paragraph 2):

“Although the current study demonstrates converging, multimodal evidence for polygenic risk, replication in additional cohorts with larger sample sizes that allow for robust cross-validation is warranted. Notably, machine-learning methods have the tendency to over-fit data and produce estimates that do not generalize to different data sets. However, alternative datasets for ALS that contain detailed genotyping and cognitive phenotyping are currently lacking and the CReATe PGB cohort represents the largest of its kind. In the absence of an alternative dataset to minimize over-fitting, we employed a bootstrapping procedure and generated a final sCCA model based on median weights across permutations rather than selecting a single “top model”. We additionally demonstrate converging, multimodal evidence for polygenic risk in independent

neuroimaging and neuropathology biomarker cohorts in an effort to provide corroboration that we are detecting a true biological signal. However, future research is necessary to determine the predictive potential and generalizability of our proposed polygenic risk score in ALS patients. We furthermore hope that this demonstration motivates the collection of additional genotyping data and longitudinal cognitive evaluation using the ECAS in additional large-scale patient cohorts.”

4. Having said this, I'm not sure how much the knowledge about such a „polygenetic“ risk score helps our further understanding of cognitive decline in ALS. Of course, it adds to the fact that cognition is not defined by a single SNP, but this is not novel.

We appreciate this reviewer’s concern for the utility of a polygenic risk score for understanding cognitive impairment in ALS. While others have demonstrated shared polygenic risk between several neurodegenerative diseases and cognitive and physical function in the UK Biobank, a large cohort of healthy individuals (Hagenaars et al., *Plos One*, 2018), to our knowledge, ours is the first study to demonstrate polygenic risk for cognitive impairment using targeted analysis of genome-wide risk variants for ALS or ALS and FTD in a symptomatic patient cohort. Notably, our results demonstrate a sparse association between several variants previously associated with case-control ALS and FTD risk and both baseline performance and rate of decline on detailed measures of cognition that largely reflect specific impairments observed in ALS. We believe that our study encourages the consideration of genetic variation in addition to phenotypic heterogeneity, including cognitive impairment, in ALS and related disorders. We address potential caveats in the Discussion (Page 22, paragraph 1):

“While our sCCA modeling selected 25 SNPs in addition to sex and *C9ORF72* mutation status and we used model-derived weights to calculate a wPRS, we are unable to determine in the current study what the collective contribution of these SNPs are to modifying cognitive phenotypes. For example, these could be additive in nature, such that increased risk allele dosage increases risk for impaired cognition, or the selected SNPs could act independently in disease modification. As often is the case, future functional studies are required to identify the mechanistic relationship between SNP associations and cognitive phenotype.”

B. Minor Comments

1. Were the C9 patients in the postmortem analysis taken out of analysis?

Like our other cohorts, the CReATe PGB and UPenn Biobank neuroimaging cohort, the autopsy cases were inclusive of individuals with a *C9ORF72* repeat expansion and our polygenic risk score includes a term for *C9ORF72* expansion status. This is described in the Results (Page 9, paragraph 2), Supplementary Table 2 and Methods (Page 34, paragraph 2).

2. Cognitive onset in the neuroimaging/autopsy cohort is defined how and how to compare this to the initial CReATe cohort, were only ALS-FTD was noted

We appreciate this reviewer’s consideration of the cognitive diagnoses of patients in the UPenn cohorts that have been collected since 1985. The cognitive diagnoses for these cohorts were made according to the revised Strong criteria (Strong et al., *ALSFTD*, 2017) using retrospective chart review by expert neurologists. We describe the cognitive diagnoses of the UPenn Biobank neuroimaging cohort as follows (Page 34, paragraph 3):

“All patients were diagnosed with ALS by a board-certified neurologist (L.E., L.M., M.G., D.I.) using revised El Escorial criteria (Brooks et al., *Amyotrophic lateral sclerosis and other motor neuron disorders : official publication of the World Federation of Neurology, Research Group on Motor Neuron Diseases*, 2000) and assessed for ALS frontotemporal

spectrum disorder using established criteria (Strong et al., *Amyotrophic Lateral Sclerosis and Frontotemporal Dementia*, 2017); those patients enrolled in research prior to 2017 were retrospectively evaluated through chart review.”

We revised the Methods for the UPenn Biobank autopsy cohort to also include this information regarding cognitive diagnoses. We now state in the Methods (Page 35, paragraph 2):

“During life, all patients were diagnosed with ALS by a board-certified neurologist (L.E., L.M., M.G., D.I.) using revised El Escorial criteria (Brooks et al., *Amyotrophic lateral sclerosis and other motor neuron disorders : official publication of the World Federation of Neurology, Research Group on Motor Neuron Diseases*, 2000) and assessed for ALS frontotemporal spectrum disorder using established criteria (Strong et al., *Amyotrophic Lateral Sclerosis and Frontotemporal Dementia*, 2017); those patients enrolled in research prior to 2017 were retrospectively evaluated through chart review.”

3. Introduction: „As many as half of patients with amyotrophic lateral sclerosis (ALS) manifest progressive decline in cognition consistent with extra-motor frontal and temporal lobe neurodegeneration...” Current understanding is that cognitive dysfunction is already present at motor disease onset and it is of current debate whether cognition also generally declines. The statement is of general and not fully proven yet!

This reviewer brings up an interesting point regarding the timing of cognitive impairment in the disease course of patients with ALS. While some have demonstrated the progression of cognitive impairment after initial motor symptom onset (e.g. Elamin et al. *Neurology*, 2013), others have also shown that cognitive dysfunction may precede motor symptom onset (e.g. Mioshi et al. *Neurology*, 2014). To more accurately reflect this current understanding, we have changed the statement in the Introduction to omit the assumption of general decline (Page 4, paragraph 1):

“A significant proportion of patients with amyotrophic lateral sclerosis (ALS) manifest impairment in cognition consistent with extra-motor frontal and temporal lobe neurodegeneration, including 14% also diagnosed with frontotemporal dementia (FTD).”

16th Oct 2020

Dear Dr. McMillan,

Thank you for the submission of your revised manuscript to EMBO Molecular Medicine. I am pleased to inform you that we will be able to accept your manuscript pending the following final amendments:

***** Reviewer's comments *****

Referee #2 (Comments on Novelty/Model System for Author):

In my review of the original submission I gave a very extensive set of suggestions and comments to which the authors responded very comprehensively

Referee #2 (Remarks for Author):

My thanks to the authors for their extensive and very clear responses to all comments. I think they have a nice body of work here. Some questions remain for follow up in future work but in my opinion the story is sufficiently complete and I have no outstanding concerns.

The authors performed the requested changes.

The authors performed the requested changes.

Corresponding Author Name: Corey T McMillan
Journal Submitted to: EMBO Molecular Medicine
Manuscript Number: EMM-2020-12595